# A single-column ocean-biogeochemistry model (GOTM-TOPAZ) version 1.0

Hyun-Chae Jung[1], Byung-Kwon Moon[1], Jieun Wie[1], Hye-Sun Park[2], Johan Lee[3], Young-Hwa Byun[3]

[1]Division of Science Education, Institute of Fusion Science, Chonbuk National University, Jeonju 54896, South Korea
[2]Cray Korea Inc., Seoul 08511, South Korea
[3]National Institute of Meteorological Sciences, Seogwipo 63568, South Korea

*Correspondence to*: Byung-Kwon Moon (moonbk@jbnu.ac.kr)

**Abstract.** Recently, Earth System Models (ESMs) have begun to consider the marine ecosystem to reduce errors in climate simulations. However, many models are unable to fully represent the ocean biology-induced climate feedback, which is due in part to significant bias in the simulated biogeochemical properties. Therefore, we developed the Generic Ocean Turbulence Model–Tracers of Phytoplankton with Allometric Zooplankton (GOTM-TOPAZ), a single-column ocean biogeochemistry model that can be used to improve ocean biogeochemical processes in ESMs. This model was developed by combining the GOTM, a single-column model that can simulate the physical environment of the ocean, and TOPAZ, a biogeochemical module. Here, the original form of TOPAZ has been modified and modularized to allow easy coupling with other physical ocean models. To demonstrate interactions between ocean physics and biogeochemical processes, the model was designed to allow ocean temperature to change due to absorption of visible light by chlorophyll in phytoplankton. We also added a module to reproduce upwelling and the air-sea gas transfer process for oxygen and carbon dioxide, which are of particular importance for marine ecosystems. The simulated variables (e.g., chlorophyll, oxygen, nitrogen, phosphorus, silicon) of GOTM-TOPAZ were evaluated by comparison against observations. The temporal variability of the observed upper-ocean (0–20 m) chlorophyll is well captured by the GOTM-TOPAZ model with a correlation coefficient of 0.51. The surface correlation coefficients between the GOTM-TOPAZ oxygen, nitrogen, phosphorus, and silicon are 0.47, 0.30, 0.16, and 0.19, respectively. We compared the GOTM-TOPAZ simulations with those from MOM-TOPAZ and found that GOTM-TOPAZ showed relatively lower correlations, which is most likely due to the limitations of the single-column model. Results also indicate that source/sink terms may contribute to the biases in the surface layer ($< 60$ m), while initial values are important for realistic simulations in the deep sea ($> 250$ m). Despite this limitation, we argue that our GOTM-TOPAZ model is a good starting point for further investigation of key biogeochemical processes and is also useful to couple complex biogeochemical processes with various oceanic global circulation models.

## 1 Introduction

Over several decades, climate researchers have accumulated significant knowledge on atmosphere-land-ocean feedback processes through various studies related to climate systems (Friedlingstein et al., 2006; Soden and Held, 2006; Dirmeyer et al., 2012; Randerson et al., 2015). With the advancement of coupled modeling techniques and an exponential increase in the number of computer resources available, climate research institutions worldwide began competing to develop earth system models (ESMs) (Dunne et al., 2012a; Dunne et al., 2012b; Jones and Sellar, 2015; Sokolov et al., 2018). ESMs are often coupled with biogeochemistry models that consider the atmosphere–ocean carbon cycle and ocean ecosystem cycles (Dunne et al. 2012b; Yool et al., 2013; Azhar et al., 2014; Stock et al., 2014; Aumont et al., 2015). Recently, reproductions of ocean ecosystems in ESMs have become very precise with the addition of physiological details, such as light or nutrient acclimation, and the division of various phytoplankton and zooplankton into functional groups (Hense et al., 2017).

The following processes are generally considered the most important in ocean biogeochemistry models: the ocean ecosystem cycle, including phytoplankton and zooplankton; the biogeochemical carbon cycle; and the biogeochemical cycle of key nutrients (P, N, Fe, and Si) (Dunne et al., 2012b; Aumont et al., 2015). These three cycles are not independent and include mutual material exchange through chemical mechanisms. There are still no accurate methodologies with which to differentiate biogeochemical variables and to represent biogeochemical processes as formulas (Sauerland et al., 2018). In other words, biogeochemical processes are reproduced in the model via parameterization that adjusts the parameters of a formula based on observations and some general parameters (e.g., maximum phytoplankton growth rate) that are adjusted until the model produces reasonable results (Sauerland et al., 2018).

Researchers have been using single-column models (SCMs) to control the parameterizations and increase their understanding of the physical processes in models. Betts and Miller (1986) suggested that SCMs were an effective tool with which to develop and control the convective scheme of an atmospheric model, while Price et al. (1986) used an ocean SCM to study the daily cycle of the mixed layer in the Pacific Ocean. An SCM allows for control of physics parameters, alongside large-scale forcing influences, and, unlike 3D models, it has a low calculation cost. Accordingly, SCMs have been viewed as essential tools with which to develop and improve numerical models (Lebassi-Habtezion and Caldwell, 2015; Hartung et al., 2018). SCM-based studies are essential for improving ocean-biogeochemical processes, which are reproduced in climate models based on column physics (Evans and Garçon, 1997; Burchard et al., 2006; Bruggenman and Bolding, 2014). Even the latest analyses of the ESMs included in the Coupled Model Intercomparison Project Phase 5 (CMIP5) show high biases and inter-model diversity in ocean biogeochemical variables (Lim et al., 2017). Therefore, a single-column form of a biogeochemistry model might be a useful tool to meet the ongoing demand for improvements in biogeochemistry models in ESMs.

The oceanic biogeochemical cycle affects not only the physical environment of the upper ocean but also that of the entire climate system, and such changes produce feedback that, in turn, alters the ocean ecosystem (Hense et al., 2017; Lim et al., 2017; Park et al., 2018). Hense et al. (2017) presented the $CO_2$ cycle, gas and particle cycle, and changes in the physical

environment of the upper ocean by chlorophyll as important climate-ocean biogeochemistry feedback loops reproduced in
ESMs that are currently available. An ESM that reproduces all three of these biological mechanisms does not exist today;
however, all of these mechanisms need to be properly reproduced in the ESMs to reduce the uncertainty in predicting future
climate change. This would allow ESMs to change in a fundamentally different way. Furthermore, there are generally time
constraints in repeated experiments using ocean general circulation models (OGCMs) and biogeochemistry models due to
their complexity and the heavy calculation required. Consequently, SCMs are crucial for applying and testing new climate-
ocean-biogeochemistry feedbacks in existing ESMs.
In this study, we developed the Generic Ocean Turbulence Model–Tracers of Phytoplankton with Allometric Zooplankton
(GOTM-TOPAZ), which is a single-column ocean-biogeochemistry model. GOTM is a one-dimensional ocean model that
focuses on reproducing statistical turbulence closures (see http://www.gotm.net); TOPAZ is an ocean-biogeochemistry
model developed by the Geophysical Fluid Dynamics Laboratory (GFDL) and coupled with the ESM2M and ESM2G
models (Dunne et al., 2012a; Dunne et al., 2012b). We modularized TOPAZ to apply external physical environmental data
while modifying it as an SCM. It was then combined with a GOTM utilizing an air-sea gas exchange for $CO_2$ and $O_2$ and
optical feedback from photosynthesis by chlorophyll. A w-advection prescription module that can reproduce upwelling was
also added to this model. To verify GOTM-TOPAZ, we selected points in the East/Japan Sea off the coast of the Korean
Peninsula upon which to conduct simulations. The results produced by the model were compared to observed data and
results from OGCMs to verify its reliability.

## 2 The Physical Ocean Model: General Ocean Turbulence Model (GOTM)

In GOTM-TOPAZ, the GOTM version 4.0 is applied to ocean physics. The physical bases of the GOTM are Reynolds-
averaged Navier–Stokes equations in a rotational coordinate system (Eqs. 1 and 2). Moreover, the temperature and salinity
equations derived using these methods are given in Eqs. 3 and 4, respectively. GOTM uses one-dimensional potential
temperature, salinity, and horizontal velocity based on these four equations, as shown below:

$$\partial_t u - \nu \partial_{zz} u + \partial_z \langle u'w' \rangle = -\frac{1}{\rho_0} \partial_x p + fv \tag{1}$$
$$\partial_t v - \nu \partial_{zz} v + \partial_z \langle v'w' \rangle = -\frac{1}{\rho_0} \partial_y p - fu \tag{2}$$
$$\partial_t T - \nu' \partial_{zz} T + \partial_z \langle w'T' \rangle = \frac{\partial_z I}{c_p \rho_0} \tag{3}$$
$$\partial_t S - \nu'' \partial_{zz} S + \partial_z \langle w'S' \rangle = \tau_R^{-1}(S_R - S). \tag{4}$$

In Eqs. (1) and (2), $u$, $v$, and $w$ represent the mean velocities in the spatial directions $x$ (eastward), $y$ (northward), and $z$
(upward), respectively; $\nu$ represents the molecular diffusivity of momentum; $\rho_0$ represents a constant reference density; $p$
represents pressure; and $f$ represents the Coriolis parameter. In Eq. (3), the temperature ($T$) equation, $\nu'$ represents the
molecular diffusivity due to heat; $c_p$ represents the heat capacity; and $I$ represents the vertical divergence of short-wave
radiation. The effect of solar radiation absorbed by seawater is included in this equation; thus, Eq. (3) is closely associated
with the radiation parameterization method. Moreover, a coupled ocean biogeochemistry model must contain an additional
short-wave absorption process associated with chlorophyll synthesis distributed throughout the upper-ocean layer (Morel and
Antoine, 1994; Cloern et al., 1995; Manizza et al., 2005; Litchman et al., 2015; Hense et al., 2017). Based on the
methodology of Manizza et al. (2005), we applied a visible light absorption process due to chlorophyll synthesis, explained
in detail in Sect 4.4, to the coupled model. Equation (4) explains the vertical distribution of salinity ($S$). In this equation, $\nu''$
represents the molecular diffusivity of salinity; $\tau_R$ represents the relaxation time scale; and $S_R$ represents the observed
salinity distribution. In other words, the terms on the right side of this equation express the "relaxation" process based on
observations. Unlike 3D models, SCMs cannot reproduce horizontal advection. Therefore, as salinity is greatly affected by
horizontal advection, it is necessary to prescribe and supplement the observed value to the simulated value with the terms on
the right side of Eq. (4) (Burchard et al., 2006). Please see Umlauf and Burchard (2003, 2005), Umlauf et al. (2005), and
Burchard et al. (2006) for further detailed information on the GOTM.

**3 The Ocean Biogeochemistry Model: Tracers of Phytoplankton with Allometric Zooplankton (TOPAZ)**
We chose TOPAZ version 2.0 to couple with the GOTM. TOPAZ simulates the nitrogen, phosphorus, iron, dissolved
oxygen, and lithogenic material cycles as well as the ocean carbon cycle while also considering zooplankton and
phytoplankton growth cycles. It divides phytoplankton into small and large groups based on size, including the group of
nitrogen-fixing diazotrophs. Consequently, TOPAZ handles a total of 30 prognostic and 11 diagnostic tracers. The local
changes in the tracers simulated in TOPAZ can be explained by the following equation:

$\partial_t C = -\nabla \cdot \vec{v}C + \nabla K\nabla C + S_C.$                       (5)

Equation (5) is an advection-diffusion equation for each state variable $C$ simulated in TOPAZ. In this equation, $\vec{v}$
represents the velocity vector calculated in the ocean model, K represents diffusivity, and $S_C$ represents the sources minus the
sinks of $C$ calculated at each point in the model. TOPAZ is received data from the ocean model in terms of the transport
tendency of the tracers associated with advection and horizontal diffusion, and it calculates vertical diffusion and source/sink
terms internally. The biological processes of TOPAZ were reproduced with a focus on phytoplankton growth, nutrient and
light limitations, the grazing process, and empirical formulas derived from observations. These are followed by the Redfield
ratio (Redfield et al., 1963), Liebig's law of the minimum (de Baar, 1994), and size considerations (large organisms feed on
smaller ones), which were used to establish the ocean ecosystem model (Dunne et al., 2012b). Please see Dunne et al. (2012b)
for further detailed information on TOPAZ.

## 4 The Ocean Biogeochemistry Coupled Model: GOTM-TOPAZ

TOPAZ was initially coupled with Modular Ocean Model 5 (MOM5), an OGCM developed by the GFDL. We separated
TOPAZ from MOM5 and constructed two modules by separating the initialization and main calculation subroutines. This
model was then modified into an SCM while adding interfaces associated with surface flux prescriptions (boundary
conditions) and initial data input.
In our new coupled model, the GOTM provided ocean physics calculations for TOPAZ, and TOPAZ relayed optical
feedback from the chlorophyll simulated according to these data to GOTM. A subroutine that calculates the optical feedback
from chlorophyll and another that prescribes the w-advection were added to GOTM-TOPAZ (see Fig. 1 for the flow
diagram). Upwelling that usually occurs along coastal areas due to wind plays a major role in changing the vertical
distribution of zooplankton and phytoplankton by supplying the surface layer with nutrient-rich intermediate water (Krezel et
al., 2005; Lips and Lips, 2010; Shin et al., 2017). We connected the w-advection module in GOTM to TOPAZ so that the
upwelling was reproduced in TOPAZ.

## 4.1 Initial Conditions

The initial data needed to run GOTM-TOPAZ can be divided into the data needed to operate the GOTM and TOPAZ models
individually. To run the GOTM, it is necessary to have the initial ocean data (temperature and salinity) and the salinity data
for the duration of the model run time. The latter are needed to relax the GOTM. For TOPAZ, initial data are needed for the
30 prognostic and 11 diagnostic tracers.

## 4.2 Boundary Conditions

Atmospheric forcing data must be prescribed in GOTM-TOPAZ because it is not coupled with an atmospheric model. The
atmospheric forcing variables needed to run the model are: 10 m u-wind; v-wind [m s$^{-1}$]; surface (2 m) air pressure [hPa];
surface (2 m) air temperature [℃]; relative humidity [%], wet bulb temperature [℃], or dew point temperature [℃]; and
cloud cover [1/10].
Values for surface or bottom fluxes for a few types of tracers must be provided to accurately simulate ocean
biogeochemical variables. TOPAZ includes processes for variables including sediment calcite cycling and the external
bottom fluxes of $O_2$, $NH_4$, $PO_4$, and alkalinity (Dunne et al., 2012b). However, it does not include a process for calculating
the atmosphere-ocean surface flux. Therefore, we added processes for calculating the surface fluxes of $O_2$, $NO_3$, $NH_4$,
alkalinity, lithogenic aluminosilicate, dissolved iron, and dissolved inorganic carbon. Of the subroutines shown in Fig. 1., the
calculation of the surface fluxes is implemented using generic_topaz_column_physics. The surface flux of $NO_3$, $NH_4$,
lithogenic aluminosilicate, and dissolved iron is prescribed using monthly average climate values, while alkalinity is
calculated from prescribed $NO_3$ dry/wet deposition values. These surface flux data are provided by the Australian Research
Council's Centre of Excellence for Climate System Science (ARCCSS; http://climate-cms.unsw.wikispaces.net/Data). The
following equation was used to calculate the air-sea gas transfer for $O_2$ and $CO_2$ (dissolved inorganic carbon):

$$F = k_w \rho([A] - [A]_{sat}) \tag{6}$$

Here, F is the upward flux of gas $A$, and $k_w$ is its gas transfer velocity, which can be calculated as a function of the
Schmidt number and wind speed at 10 m (Wanninkhof, 1992). $\rho$ is the density of surface seawater, $[A]$ is the concentration
[$\mu$mol $kg^{-1}$] of gas A at the surface of the ocean, and $[A]_{sat}$ is the corresponding saturation concentration of gas A in
equilibrium with a water vapor-saturated atmosphere at total atmospheric pressure (Najjar and Orr, 1998). $[A]$ is predicted by
the model. Please see Najjar and Orr (1998) for further detailed information related to Eq. (6).

**4.3 Ocean Physics**
The GOTM simulates the physics of oceanic environments based on Eqs. (1)–(4). In the coupled model, the GOTM relays
the following simulated one-dimensional ocean physical variables to the TOPAZ module at each time step: potential
temperature [°C]; salinity [psu]; thermal diffusion coefficient [$m^2\ sec^{-1}$]; density [kg $m^{-3}$]; thickness [m]; mixed layer
thickness [m]; and radiation [w $m^{-2}$].

**4.4 Optical Feedback**
As explained in Sect. 2, the photosynthesis of chlorophyll distributed throughout the upper ocean is known to have physical
effects. Manizza et al. (2005) used satellite observation data and OGCMs to conduct a study of changes in ocean irradiance
due to the absorption of visible light by chlorophyll. We used their methodology to apply the optical feedback from
chlorophyll on GOTM-TOPAZ in the following manner:

$$k_\lambda = k_{sw(\lambda)} + \chi_{(\lambda)} \cdot [chl]^{e(\lambda)} \tag{7}$$
$$I_{IR} = I_0 \cdot 0.58 \tag{8}$$
$$I_{VIS} = I_0 \cdot 0.42 \tag{9}$$
$I_{RED} = I_{BLUE} = \frac{I_{VIS}}{2}$                    (10)
$I_{(z)} = I_{IR} \cdot e^{-k_{IR}z} + I_{RED(z-1)} \cdot e^{-k_{(r)}\Delta z} + I_{BLUE(z-1)} \cdot e^{-k_{(b)}\Delta z}.$          (11)

In these equations, visible light was divided into red and blue/green bands in accordance with Manizza et al. (2005). In Eq.
(7), λ represents the wavelength of these bands and $k_{sw(\lambda)}$ represents the light attenuation coefficient of optically pure
seawater, which has values of $0.225 \text{ m}^{-1}$ and $0.0232 \text{ m}^{-1}$, respectively, in red and blue/green bands. In these bands, the
values of the pigment adsorption $\chi_{(\lambda)}$ are 0.037 and 0.074 $\text{m}^{-2}$ mg Chl $\text{m}^{-3}$, respectively; $e_{(\lambda)}$, the power law for absorption,
has values of 0.629 and 0.674 [no units], respectively. Moreover, [chl] represents the concentration of chlorophyll in
mg Chl $\text{m}^{-3}$.
Infrared light ($I_{IR}$) and visible light ($I_{VIS}$) that reach mean open ocean conditions are set in Eqs. (8) and (9), respectively,
by default. However, GOTM-TOPAZ can change the light extinction method by modifying the namelist in the GOTM (see
http://www.gotm.net) and this can also be used to change the coefficients of $I_{IR}$ and $I_{VIS}$. The total irradiance of the red and
blue/green bands that reach the ocean surface is represented in Eq. (10). Ultimately, the irradiance of visible light transmitted
at each vertical level (z) can be calculated in GOTM-TOPAZ using Eq. (11). Moreover, the sum of the second and third
terms on the right side of Eq. (11) represents photosynthetically active radiation (PAR) and is used in TOPAZ to calculate
the growth rate of phytoplankton groups.

**4.5 w-advection**
As mentioned at the beginning of Sect. 4, the upwelling phenomenon generated by coastal winds is known to affect
phytoplankton growth by supplying nutrient-rich intermediate water to the upper ocean. The GOTM is already designed to
allow users to prescribe w-advection to experiments. Therefore, we linked the subroutines of the GOTM that are related to
w-advection to TOPAZ, so GOTM-TOPAZ users can study the impact of upwelling on the biogeochemical environment of
the ocean. Users can prescribe vertical advection as a constant or input the velocities by time and depth in ASCII format to
reproduce the desired form of vertical motions. Please refer to the GOTM homepage (http://www.gotm.net) and Burchard et
al. (2006) for further technical details and numerical analysis of the w-advection in GOTM.

**5 Experimental Setup**
The East/Japan Sea is unique, with its steep topography and three large, deep, and semi-enclosed basins. Moreover, it is
somewhat isolated from other major oceans, connects to the Pacific Ocean through a narrow strait, and is sometimes referred
to as a miniature ocean since it contains a double gyre and experiences various oceanic phenomena (Ichiye, 1984). The high-

temperature, high-salinity Tsushima Warm Current (TWC) introduced through the Korea Strait is divided into two main branches: the nearshore branch, which flows northeastward along the Japanese coast, and the East Korea Warm Current (EKWC), which flows northward along the Korean coast (Uda, 1934; Tanioka, 1968; Moriyasu, 1972) (Fig. 2). Apart from these two main branches, there is another that exists offshore of the first branch, but it is not present all year (Shimomura and Miyata, 1957; Kawabe, 1982). To the north, the North Korea Cold Current (NKCC) flows southward along the Korean coast. Furthermore, the 200−400 m East Sea Intermediate Water (ESIW) is known for its high concentration of dissolved oxygen and the appearance of a salinity-minimum layer (Kim and Chung, 1984; Kim and Kim, 1999). The East/Japan Sea is divided into warm and cold regions relative to the 40° N parallel, and, since the current pattern and characteristics of the East/Japan Sea vary spatially and seasonally, this region is very important to oceanographic studies. This region is also considered important for biogeochemical research (Joo et al., 2014; Kim et al., 2016; Shin et al., 2017) for the following reasons: the nutrient-rich seawater that flows along the southern coast of the Korean Peninsula due to inflow from the Nakdong River, which is located at its southeastern end; the influence of a strong southerly wind during the summer, which causes upwelling off the coast of the East/Japan Sea; and the transport of this nutrient- and chlorophyll-rich seawater near Ulleungdo Island by the EKWC. We selected three points that have features typical of the East/Japan Sea and for which observation data suitable to use for verification exist (Fig. 2): point 107, where the EKWC and NKCC meet (130.0° E, 38.0° N); point 104, which is an important location along the EKWC (131.3° E, 37.1° N); and point 102, which is in the middle of a warm eddy created as the EKWC moves north (130.6° E, 36.1° N). As noted previously, these points are in regions with strong advection and thus may not be suitable for testing GOTM-TOPAZ, which is an SCM. However, since the results obtained using GOTM-TOPAZ were significant when compared to the observations, we think that this shows that it is possible to perform sensitivity experiments using GOTM-TOPAZ at several kinds of locations.

The observed data, such as seawater temperature and salinity, were used to initialize and relax vertical structures in the GOTM throughout the simulation. These data were provided by the National Institute of Fisheries Science (NIFS; http://www.nifs.go.kr/kodc). The water temperature and salinity data from the NIFS were measured at 15-m intervals at depths of 0 m to 500 m. They were measured once in February, April, June, August, October, and December every year beginning in 1961. For the initial data on prognostic/diagnostic tracers in TOPAZ, we used the data provided by ARCCSS for use with MOM5 (http://climate-cms.unsw.wikispaces.net/Data). These initial tracer data were interpolated for each location, and a spin-up was applied over 14 years for use in the experiments. For atmospheric forcing data, we input 0.75° ERA-Interim reanalysis data provided by the European Centre for Medium Range Weather Forecasts (Dee et al., 2011). We applied global data to our model by interpolating the latitude and longitude values of the test points.

We used the monthly average of observed seawater temperature and salinity data from the analysis fields in the EN.4.2.1, provided by the Hadley Centre at the Met Office (Good et al., 2013), to verify the results from GOTM-TOPAZ following the adjusted method in Gouretski and Reseghetti (2010). With respect to chlorophyll, we compared the results simulated by the model using observational data with a resolution of 9 km gathered by the NASA Goddard Space Flight Center's Sea-Viewing Wide Field-of-View Sensor (SeaWiFS) from October 1997 to December 2007 (McClain et al., 1998). The results of

simulations of dissolved oxygen and nutrients such as nitrogen, phosphorus, and silicon were tested using observational data
from the NIFS; these data were measured once every year, in February, April, June, August, October, and December, at
depths of 0, 20, 50, and 100 m. Specific measurement dates and times were not fixed, so we viewed the measurement data as
values that represented each month and used them to verify the model. Data from a model that operated MOM5, the Sea-Ice
Simulator, and TOPAZ together (MOM) were used for comparative analysis. MOM was operated using CORE-II forcing
data (Large and Yeager, 2009) from 1950 to 2008. We also used data from the Surface Ocean $CO_2$ Atlas (SOCAT) (Bakker
et al., 2016) from the analysis period to verify the $CO_2$ air-sea gas flux in TOPAZ. The time periods for which SOCAT
observational data exist for point 102 are April 2001, January 2005, November 2008, and December 2008. For points 104
and 107, the time period is April 2001. Finally, we performed a spin-up for 14 years on the initial data at each point and
analyzed the results of operating GOTM-TOPAZ from 1999 to 2008.

## 260  6 Results

Figure 3 shows the results of the GOTM-TOPAZ simulation and observational data (EN.4.2.1) as vertical distributions of the
water column over time. The vertical distributions of salinity at all points are well simulated and are comparable to the
observations, although this could also be because relaxation was applied. The water temperature at point 107, as simulated
by GOTM-TOPAZ, showed a cold bias in the upper layer at a depth of around 120 m (Fig. 3a). This appears to be the effect
of large-scale forcing (from the EKWC) that GOTM-TOPAZ could not resolve. Similar differences in water temperature
also appeared at points 104 and 102 (Fig. 3b and 3c). Observational results showed that the water temperature was
particularly affected by the ESIW, a finding that did not appear in the GOTM-TOPAZ results. It was determined that since
GOTM-TOPAZ could not reproduce advection from the ESIW, there were differences (warm bias) in the vertical water
temperature distributions near depths of 200 m compared to the observational results at all points (Fig. 3).
We used SeaWiFS data to measure chlorophyll concentrations using light reflected from the ocean surface and thus
verified the results simulated by GOTM-TOPAZ. However, part of the reflected light reaches the satellite from the mixed
layer below the ocean surface due to a backscattering effect (Jochum et al., 2009; Park et al., 2013). Therefore, we compared
chlorophyll anomalies averaged up to 20 m in the data from each model and chlorophyll from SeaWiFS. The mean
chlorophyll concentration at depths of 0–20 m, as simulated by GOTM-TOPAZ and MOM, had similar seasonal variabilities
at point 107; their correlation coefficients versus the observational data were 0.53 and 0.60, respectively, which is
statistically significant ($p < 0.001$) (Fig. 4a). At points 104 and 102, these correlation coefficients of GOTM-TOPAZ versus
the observational data were 0.25 ($p < 0.01$) (Fig. 4c) and 0.32 ($p < 0.001$) (Fig. 4e), respectively. In the case in which the
maximum concentration of chlorophyll at all points occurred annually on the surface layer, GOTM-TOPAZ showed smaller
errors against the observational results than did MOM (Fig. 4a, 4c, and 4e).
Phytoplankton in the East/Japan Sea are generally present in the highest concentrations at depths of around 10−60 m (Rho
et al., 2012). Therefore, we averaged chlorophyll concentrations from 20–80 m to verify the model results (Fig. 4b, 4d and
4f). However, since observational data for chlorophyll in the subsurface layer (~20–80 m) were unavailable, the MOM and
GOTM-TOPAZ results were compared instead. There were slight differences in the scale of the minimum and maximum
concentrations of chlorophyll in the subsurface layer at point 107, but the two models had a correlation coefficient of 0.59 (p
< 0.01) and a similar seasonal variability (Fig. 4b). At points 104 and 102, the GOTM-TOPAZ chlorophyll results had a
slightly lower correlation coefficient against the observational data than MOM did, but its seasonal variability was similar to
that of the observation data and the results from MOM (Fig. 4d and 4f). However, when compared to the results from MOM,
the time series of the chlorophyll anomaly in the ocean surface and subsurface layers simulated by GOTM-TOPAZ appear to
show a time shift (Fig. 4). In the TOPAZ module in MOM, the transport tendencies of each tracer were calculated in the
ocean model; however, this process was not carried out in GOTM-TOPAZ. In addition, MOM and GOTM-TOPAZ are not
only just different models of the marine physical environment; the atmospheric forcing data they each use are also different.
Therefore, there are complex reasons for the differences in the results of the two models, and further detailed experiments
and analysis are required.
We evaluated the performance of GOTM-TOPAZ in terms of simulations of dissolved oxygen, nitrogen, phosphorus, and
silicon. The sea surface dissolved oxygen at point 107 simulated by GOTM-TOPAZ and MOM had correlation coefficients
of 0.47 ($p < 0.001$) and 0.50 ($p < 0.001$), respectively, versus the observed data (Fig. 5a). The GOTM-TOPAZ correlation
coefficient versus the observed data was 0.31 ($p < 0.001$) for nitrogen, 0.16 ($p < 0.10$) for phosphorus, and 0.19 ($p < 0.05$)
for silicon; these were lower than the correlation coefficients between MOM and the observed data (0.36, 0.24, and 0.33,
respectively; $p < 0.001$). However, GOTM-TOPAZ seemed to depict the seasonal variability of nutrients at the sea surface
well (Fig. 5b–d). At point 104, the GOTM-TOPAZ correlation coefficient was 0.37 ($p < 0.001$) for dissolved oxygen, 0.54 (p
< 0.001) for nitrogen, 0.2 ($p < 0.05$) for phosphorus, and 0.1 (statistically non-significant) for silicon (Fig. 6). For point 102,
the GOTM-TOPAZ correlation coefficient was 0.59 ($p < 0.001$) for dissolved oxygen, 0.24 ($p < 0.01$) for nitrogen, 0.09
(statistically non-significant) for phosphorus, and 0.2 ($p < 0.01$) for silicon (Fig. 7). In these two points, GOTM-TOPAZ
showed values for surface dissolved oxygen and nutrients with seasonal variabilities that were similar to those of the
observed data and the data from MOM (Figs. 6–7).
Figures 8–10 show a comparison of the vertical profiles of dissolved oxygen, nitrogen, phosphorus, and silicon averaged
for February, August, and the entire period from 1999 to 2008 at points 107, 104, and 102. Mixing in the upper ocean occurs
actively during winter due to strong winds, and GOTM-TOPAZ simulated dissolved oxygen (surface to 250 m) and nitrogen
(surface to 100 m) concentrations well during that season (Figs. 8–10a). However, for phosphorus and silicon at the same
depths, there was a difference between the GOTM-TOPAZ results and the observational data. In the case of all points, the
concentrations of nitrogen, phosphorus, and silicon simulated by GOTM-TOPAZ from the surface to 60 m decreased during
August, and these concentrations were clearly distinguishable from each depth due to strong stratification in the summer
(Figs. 8–10b). These stratifications appeared in the observational data. During this season, the oxygen concentration
simulated by GOTM-TOPAZ, increased sharply from depths of 20−60 m at points 107, 104, and 102 (Figs. 8–10b). This
seems to have been caused by the creation of oxygen from photosynthesis by phytoplankton. However, a highly concentrated
dissolved oxygen concentration is not apparent in the observational data, because the warm water, which is characterized by
low dissolved oxygen, is transported by the EKWC during the summer season (Rho et al., 2012). The concentrations of
dissolved oxygen from 80−250 m at point 107 were similar in both the results from GOTM-TOPAZ and in the 10-year
observational data (Fig. 8c). However, the differences increased beyond depths of 250 m. Nonetheless, the results
demonstrated that dissolved oxygen at 80−250 m, nitrogen, and phosphorus (but not silicon) are well simulated over 10 years
using GOTM-TOPAZ (Fig. 8c). The vertical distributions of dissolved oxygen and nutrients at points 104 and 102 as
simulated by GOTM-TOPAZ over the same time period also showed similar patterns as those at point 107 (Figs. 9–10).
In addition, the magnitudes of the source and sink terms of GOTM-TOPAZ were analyzed. When TOPAZ was
implemented three-dimensionally by being coupled with MOM, the concentration of tracers was calculated through
advection-diffusion processes as well as source/sink processes. On the other hand, in the case of GOTM-TOPAZ, which is
an SCM, it determined the tendency of state variables through vertical diffusion and source and sink terms without
considering advection and horizontal diffusion. At every point, the bias of dissolved oxygen seemed to be larger in summer
than in winter, where the vertical diffusion is stronger. Since there was a bias also in the deep sea (< 250 m), we focused on
source and sink terms rather than on vertical diffusion. Figures 11–13 show 10-year (1999–2008) average source and sink
terms of nutrients (nitrate, phosphate, silicate) and dissolved oxygen. The production of dissolved oxygen is attributable to
nitrate, ammonia, and nitrogen fixation, while its loss occurs in the production of $NH_4$ from non-sinking particles, sinking
particles, and dissolved organic matter and nitrification. The production of nitrate is caused by nitrification, and its loss is
determined by denitrification and uptake by phytoplankton. In the phosphate and silicate, the production is attributable to
dissolved organic matter and particles, and the loss is determined by uptake due to phytoplankton (Dunne et al., 2012b).
As shown in Figures 11–13, the source and sink of dissolved oxygen and nutrients occurred mainly in the surface layer (<
60 m), and their influence seemed to be negligible at deeper depths. The source of dissolved oxygen was remarkable in the
surface layer during summer, because phytoplankton flourishes in summer. This pattern was commonly observed at all three
points. The surface layer of point 102, which is the southernmost point, showed more production (consumption) of dissolved
oxygen (nutrients) than did the other points in winter. Being located at the southernmost location, point 102 was greatly
affected by the warm current (EKWC), which resulted in flourishing phytoplankton. However, even at this point, the source
and sink of both the dissolved oxygen and nutrients made few contributions at 250 m or deeper.
Accordingly, it could be inferred that the simulation of biogeochemical variables in the deep sea (< 250 m) would be more
affected by initial values than by source/sink. In order to verify this assumption, the model was simulated by setting the
initial data as the observations. The results indicated that the bias of dissolved oxygen was significantly reduced in the deep
sea (Fig. 14). This result indicates that tracers simulated by GOTM-TOPAZ greatly depend on source/sink processes in the
surface layer (< 60 m) and are sensitive to initial values in the deep sea.
Finally, to verify the air-sea gas exchange simulated by GOTM-TOPAZ, we compared the monthly average sea surface
$CO_2$ concentrations in the model and in SOCAT. The correlation coefficient between the sea surface $CO_2$ concentration
simulated by GOTM-TOPAZ and the observational data was 0.94 (Fig. 15). However, there were no more than six months
for which the observational values existed at all points; therefore, this is a statistically insignificant value.

## 352 7 Discussion

In this paper, we explain the major models that comprise GOTM-TOPAZ and the biological-physical feedback loop that they
reproduce. In addition, we compiled data from three points of scientific importance in the East/Japan Sea, near the Korean
Peninsula and analyzed the results of operating GOTM-TOPAZ for a decade (~1999–2008). We compared ocean water
temperatures, salinity, and biogeochemical variables such as chlorophyll, dissolved oxygen, nitrogen, phosphorus, and
silicon concentrations against the observational data and output from the OGCM to evaluate the performance of GOTM-
TOPAZ. The results showed that GOTM-TOPAZ had lower correlation coefficients than did OGCM but that it simulated
seasonal variability in a similar manner overall. In addition, we analyzed the magnitudes of the source/sink terms for
dissolved oxygen and nutrients, which were simulated by GOTM-TOPAZ. This analysis revealed the characteristics of the
model and the cause of the bias, which was shown in the vertical profile of dissolved oxygen. Consequently, GOTM-TOPAZ
is mainly affected by source/sink terms in the surface layer (< 60 m) and is sensitive to initial values in the deep sea (> 250
m). Future users of GOTM-TOPAZ need to consider such characteristics when designing an experiment.
The SCM (1D model) includes important physical processes and has a much lower computation cost than do the 3D
models; this means that a variety of experiments can be performed repeatedly. With this advantage, 1D models can be useful
to track mechanisms that are difficult to understand using 3D models. We believe that TOPAZ, in particular, can be used to
obtain insights on the interactions between the chemical makeup and organisms in the ocean because it accounts for complex
biogeochemical mechanisms. In addition, the key processes which are studied via TOPAZ can later be implemented into 3D
models.
A variety of single-column ocean biogeochemical models have already been developed. However, GOTM-TOPAZ
includes complex biogeochemical processes and models over 30 kinds of tracers; the other models, which have only simple
structures, do not (Dunne et al., 2012b). Furthermore, GOTM-TOPAZ considers the gas transfer caused by changes in the
atmosphere and the physical environment of the ocean, depicting the deposition of dissolved iron, lithogenic aluminosilicate,
$NH_4$, and $NO_3$ due to aerosols. We believe that the sophistication of TOPAZ provides researchers with the opportunity to
perform a variety of experiments.
For example, aerosol concentrations are continuously increasing over the East Asia region and are known to affect
precipitation and atmospheric circulation. Thus, there is a possibility that aerosols affect oceanic biogeochemical processes
as deposition occurs into the ocean, and this cannot be ignored. A variety of numerical experiments are necessary to
understand this process, but they are difficult to perform using 3D models due to limitations in computing resources.
However, as previously noted, GOTM-TOPAZ is fast; as such, it is useful for understanding the biogeochemical changes
that occur in the ocean when the concentration of aerosols or $CO_2$ in the atmosphere changes. In addition, recent studies have
reported that the distribution of fisheries is changing due to changes in phytoplankton size structure caused by the upwelling
intensity on the coast of the East/Japan Sea (Shin et al., 2017). The TOPAZ phytoplankton are divided into two types
depending on their size, which should prove to be useful in this type of future research.
In addition, GOTM-TOPAZ can be used in studies on feedback mechanisms in the biogeochemical and physical
environment of the ocean. Sonntag and Hense (2011) used a simple biogeochemistry model linked to GOTM (GOTM-BIO)
to analyze the effects of phytoplankton on the physical environment of the upper ocean. The feedback from cyanobacteria,
particularly during surface blooms that cause changes in ocean surface albedo, the solar light absorption rate, and the
momentum relayed to the ocean by wind were applied to the model during the experiment. Sonntag and Hense (2011)
provided us a better understanding of the needs and direction to focus on with GOTM-TOPAZ, and we plan to apply various
climate-ocean biogeochemistry feedback mechanisms to it in future research. We also plan to evolve GOTM-TOPAZ into a
single ESM by coupling an atmospheric SCM and a model that reproduces atmospheric chemical mechanisms with GOTM-
TOPAZ.
We separated TOPAZ from MOM and constructed a model with separate initiation and column physics modules, thus
introducing the possibility of more easily coupling it with various other ocean models in the future. We are currently
conducting a study on coupling TOPAZ with the Nucleus for European Modelling of the Ocean (NEMO), another OGCM
that is already coupled with other biogeochemistry models, such as the MEDUSA (Yool et al., 2013) and the PISCES
(Aumont et al., 2015). If NEMO and TOPAZ can be coupled successfully, a comparative analysis of the simulation results
from the each biogeochemistry model might provide the driving force for improving the modelling of physical processes
associated with ocean-biogeochemistry.

**Code and data availability:**
The GOTM-TOPAZ software is based on GOTM version 4 and MOM version 5, both available for download from their
respective distribution sites (https://gotm.net, https://www.gfdl.noaa.gov/). GOTM-TOPAZ is freely available at
https://doi.org/10.5281/zenodo.1405270.

**Author contribution:**

H.C.J. and B.K.M. drafted the paper, performed the experiments, and were primarily responsible for developing GOTM-TOPAZ. J.W., H.S.P., J.L., and Y.H.B. contributed to code debugging and writing the paper.

**Competing interests:**

The authors declare that they have no conflicts of interest.

**Acknowledgements:**

We would like to thank the GOTM and MOM communities for their support. In addition, we would like to thank the European Centre for Medium Range Weather Forecasts for providing ERA-Interim data and the Hadley Centre at the Met Office for providing the EN4 datasets. In addition, we would like to thank the National Institute of Fisheries Science for providing ocean observation data and the NASA Goddard Space Flight Center for providing SeaWiFS datasets. We also thank D.H. Kim at Kongju National University of Korea for providing some advice during this research. We appreciate J.H. Choi and H.K. Kim at Chonbuk National University of Korea for their helpful discussion and comments. This work was funded by the Korea Meteorological Administration Research and Development Program under Grant KMI (KMI2018-03513).

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

**Table 1: List of abbreviations.**

| Abbreviation | Full form |
| --- | --- |
| ESM | Earth System Model |
| SCM | Single Column Model |
| OGCM | Ocean Global Circulation Models |
| CMIP5 | Coupled Model Intercomparison Project 5 (fifth phase) |
| GFDL | Geophysical Fluid Dynamics Laboratory |
| ARCCSS | Australian Research Council Centre of Excellence for Climate System Science |
| NIFS | National Institute of Fisheries Science |
| ESM2M | Earth System Model version 2, with Modular Ocean Model Version 4.1 |
| ESM2G | Earth System Model version 2, with General Ocean Layer Dynamics |
| ECMWF | European Centre for Medium-Range Weather Forecasts |
| GOTM | General Ocean Turbulence Model |
| TOPAZ | Tracers of Phytoplankton with Allometric Zooplankton |
| MOM5 | Modular Ocean Model version 5 |
| NEMO | Nucleus for European Modelling of the Ocean |
| MEDUSA | Model of Ecosystem Dynamics, Nutrients Utilization, Sequestration and Acidification |
| PISCES | Pelagic Interactions Scheme for Carbon and Ecosystem Studies |
| SOCAT | Surface Ocean $CO_2$ Atlas |
| SeaWiFS | Sea-viewing Wide Field-of-view Sensor |
| CORE-II | Coordinated Ocean-ice Reference Experiments II |
| PAR | Photosynthetically Active Radiation |
| TWC | Tsushima Warm Current |
| EKWC | East Korea Warm Current |
| NKCC | North Korea Cold Current |
| NB | Nearshore Branch |
| OB | Offshore Branch |
| ESIW | East Sea Intermediate Water |


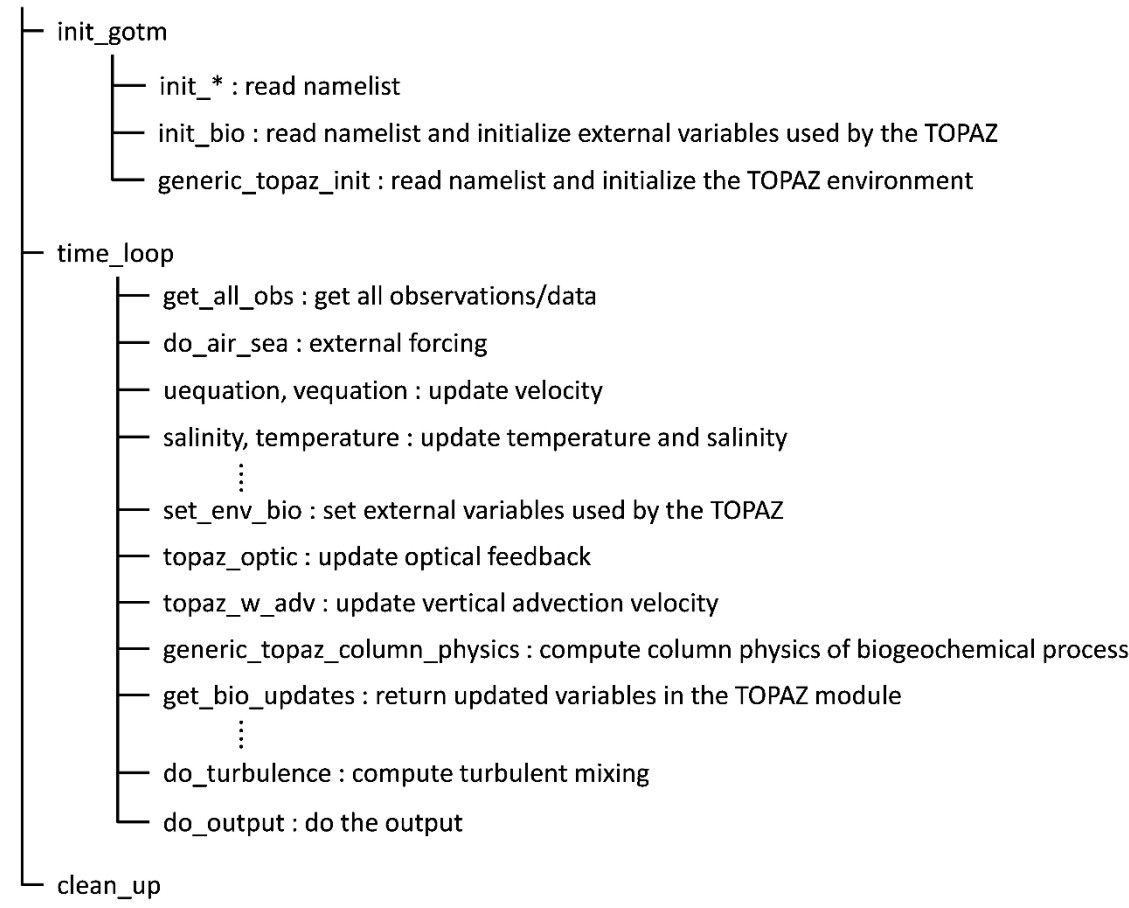


**Figure 1. Flow diagram of the Fortran subroutines comprising the Generic Ocean Turbulence Model–Tracers of Phytoplankton**
**with Allometric Zooplankton.**




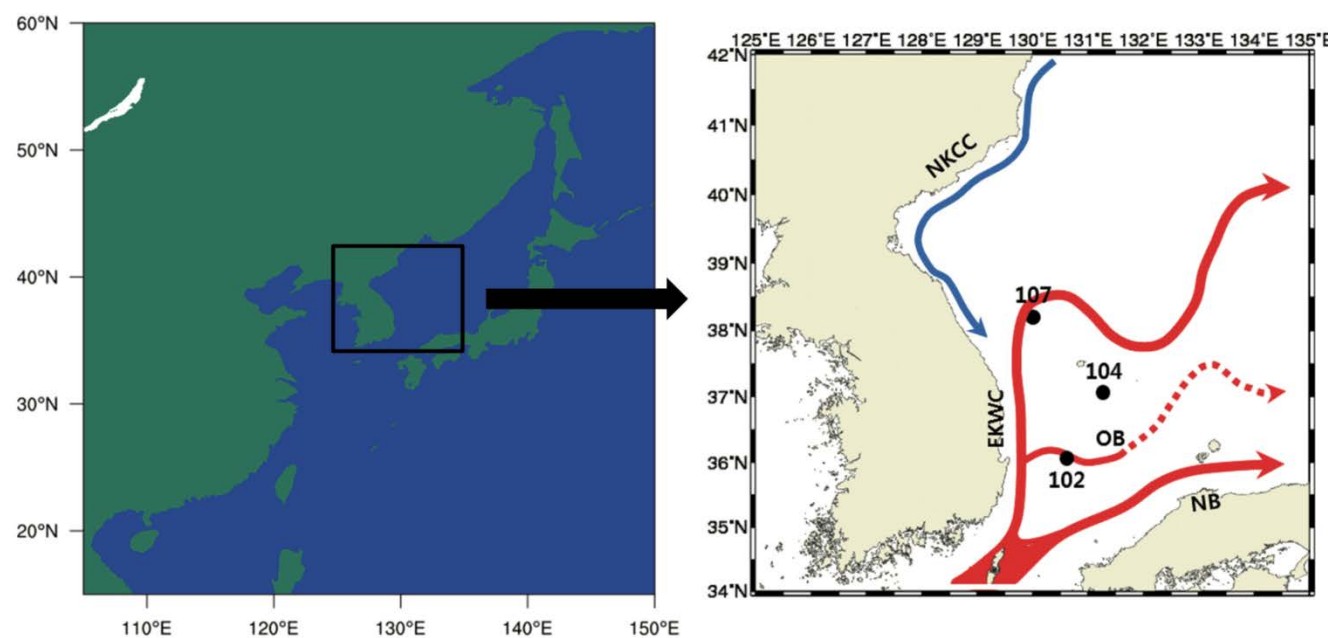

**Figure 2. Location of points (107, 104, 102) in the East/Japan Sea and flow of the nearby North Korea Cold Current (NKCC), East**
**Korea Warm Current (EKWC), Offshore Branch (OB) of the Tsushima Warm Current, and the Nearshore Branch (NB) of the**
**Tsushima Warm Current.**



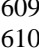

**Figure 3. Comparison of the vertical distribution for water temperature [℃], salinity [psu], and the difference (GOTM-TOPAZ**
**minus the observations) at points (a) 107, (b) 104, and (c) 102 for the 10-year period (1999−2008).**



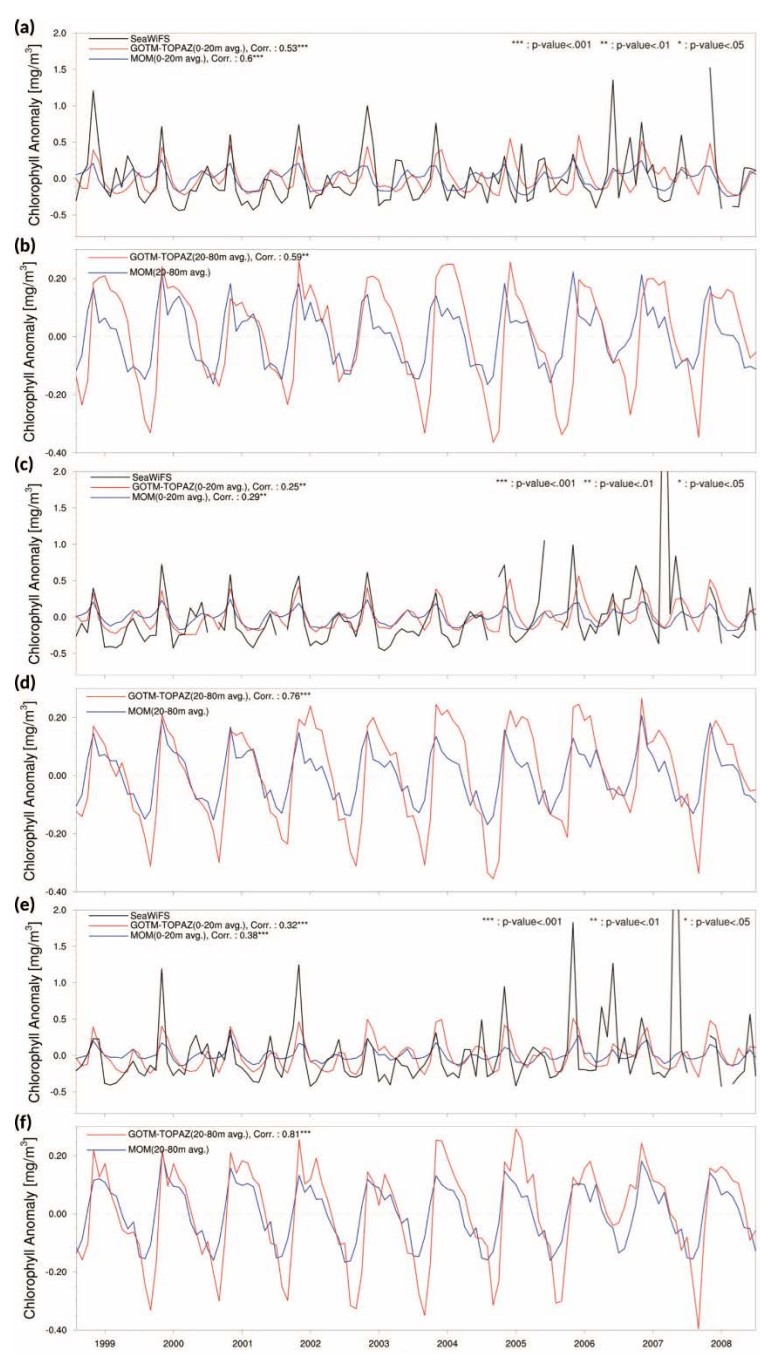


**Figure 4. Chlorophyll anomaly time series and correlation values for observational data (black lines), MOM5_SIS_TOPAZ results**
**(blue lines), and GOTM-TOPAZ results (red lines) for the 10-year period 1999−2008. (a), (c), and (e) are the mean values at depths**
**≥ 20 m and the correlations between the observations and each model at points 107, 104, and 102, respectively. (b), (d), and (f) are**
**the mean values at depths of 20−80 m and the correlation between the two models at points 107, 104, and 102, respectively.**

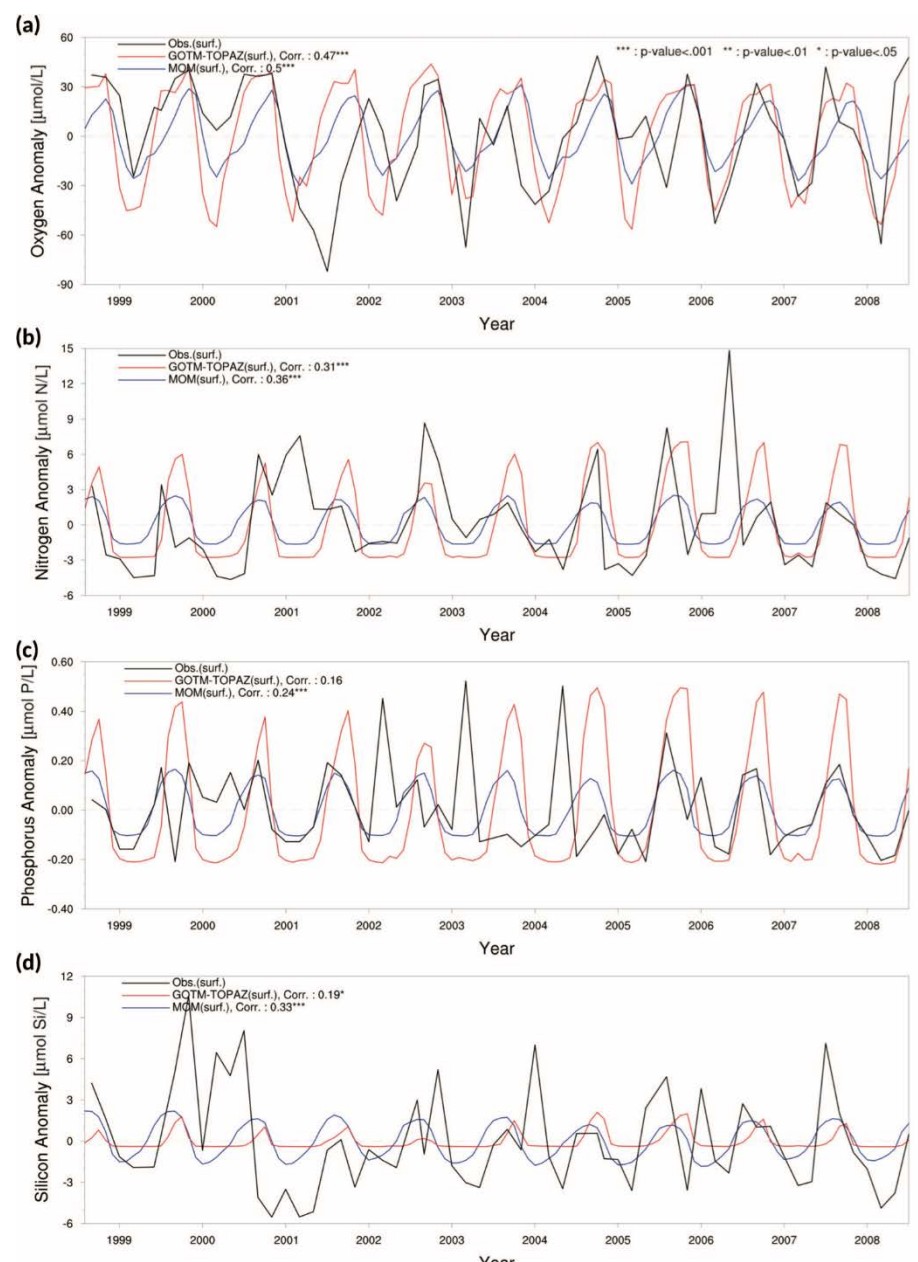


**Figure 5. Anomaly time series and correlation values from observational data (black lines), MOM results (blue lines), and GOTM-TOPAZ results (red lines) for concentrations of (a) dissolved oxygen, (b) nitrogen, (c) phosphorus, and (d) silicon at point 107 for the 10-year period 1999−2008; in this figure, nitrogen, phosphorus, and silicon include NO₃, PO₄, and SIO₄, respectively.**


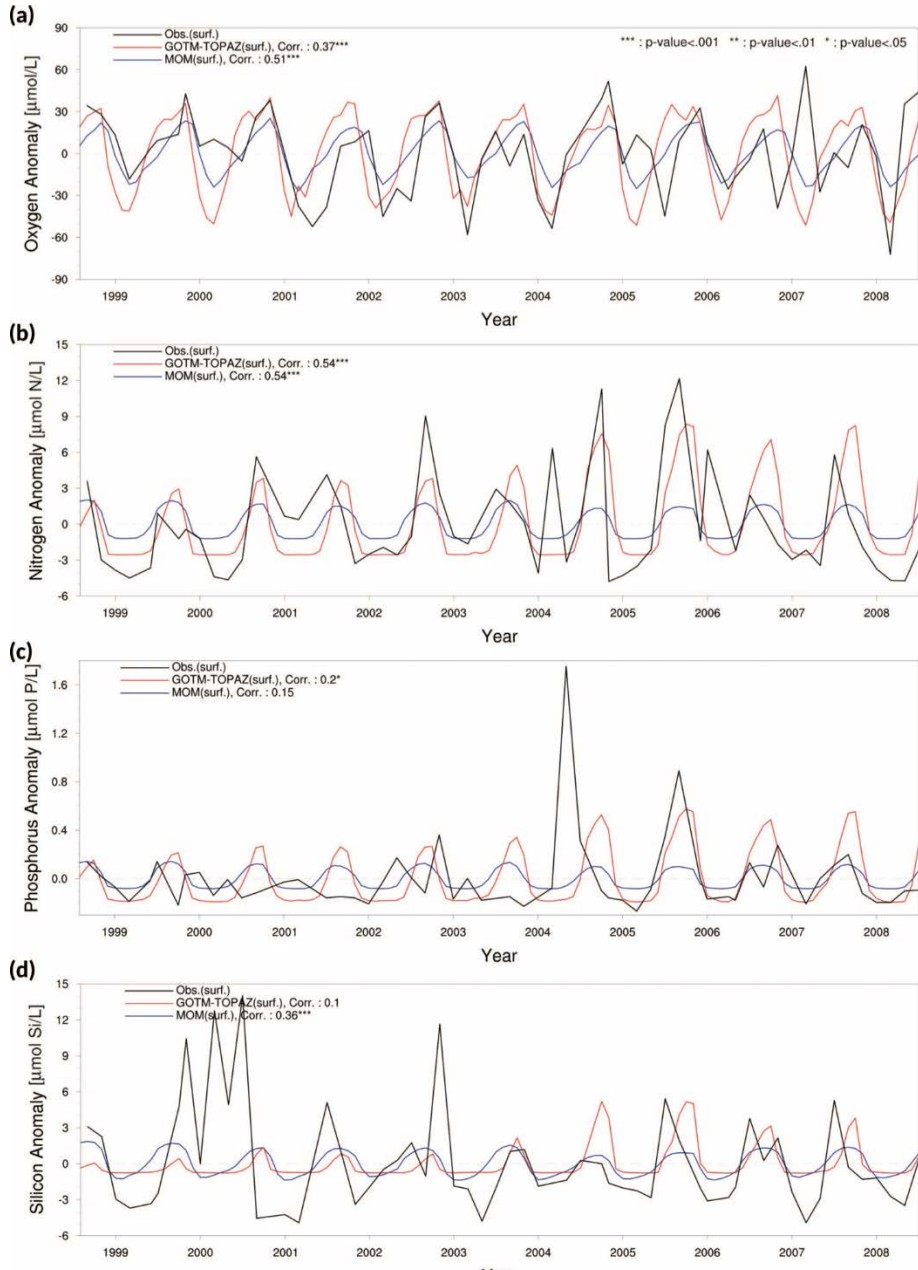

**Figure 6. Anomaly time series and correlation values from observational data (black lines), MOM results (blue lines), and GOTM-**
**TOPAZ results (red lines) for concentrations of (a) dissolved oxygen, (b) nitrogen, (c) phosphorus, and (d) silicon at point 104 for**
**the 10-year period 1999−2008; in this figure, nitrogen, phosphorus, and silicon include NO$_3$, PO$_4$, and SIO$_4$, respectively.**

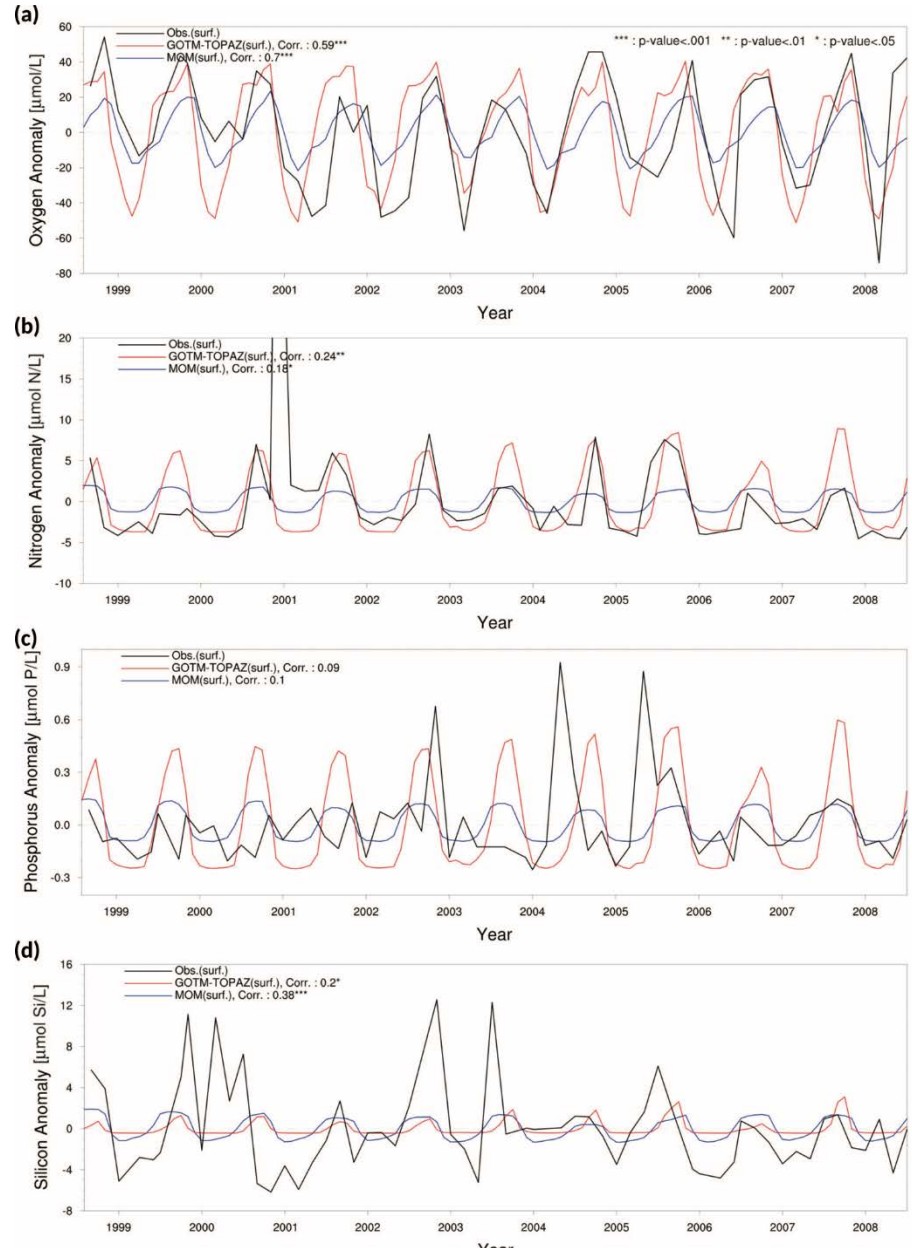


**Figure 7. Anomaly time series and correlation values from observational data (black lines), MOM results (blue lines), and GOTM-**
**TOPAZ results (red lines) for concentrations of (a) dissolved oxygen, (b) nitrogen, (c) phosphorus, and (d) silicon at point 102 for**
**the 10-year period 1999−2008; in this figure, nitrogen, phosphorus, and silicon include NO₃, PO₄, and SIO₄, respectively.**

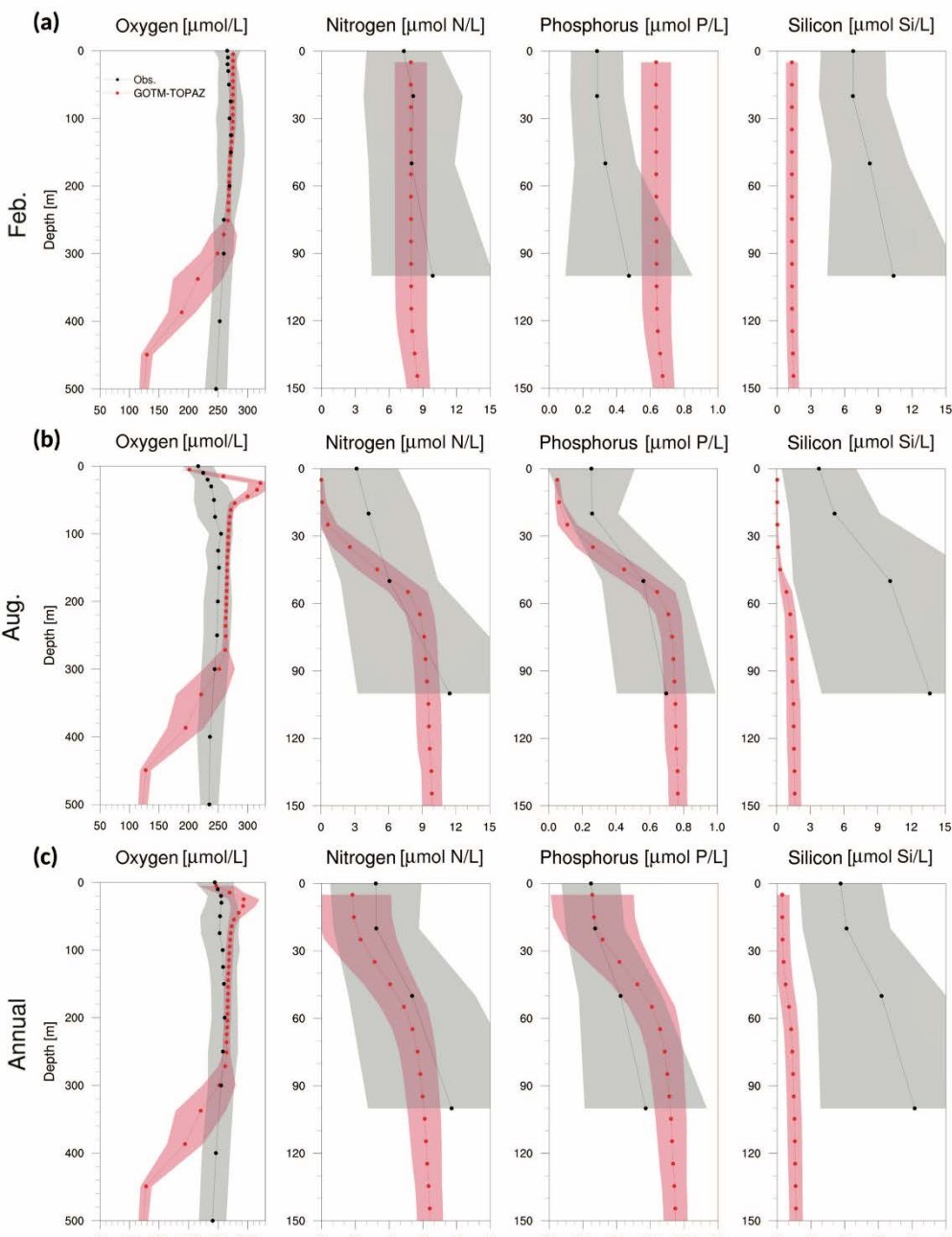


**Figure 8. Vertical profiles from observational data (black dots) and GOTM-TOPAZ results (red dots) at point 107 for**
**concentrations of dissolved oxygen, nitrogen, phosphorus, and silicon averaged from 1999–2008; (a) for February; (b) for August;**
**and (c) annually. The shaded areas represent 1 sigma. In this figure, nitrogen, phosphorus, and silicon include NO$_3$, PO$_4$, and SIO$_4$,**
**respectively.**




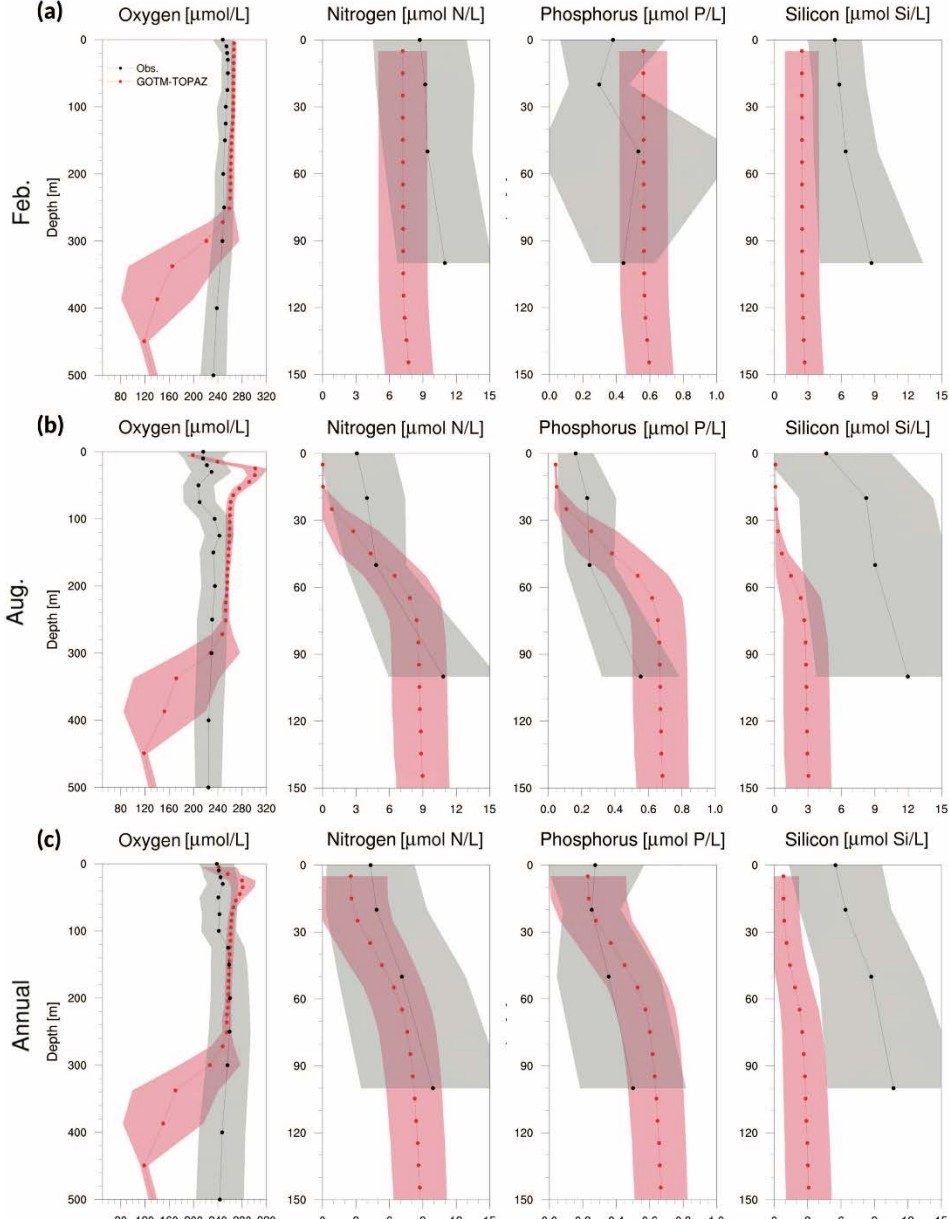


Figure 9. Vertical profiles from observational data (black dots) and GOTM-TOPAZ results (red dots) at point 104 for concentrations of dissolved oxygen, nitrogen, phosphorus, and silicon averaged from 1999–2008; (a) for February; (b) for August; and (c) annually. The shaded areas represent 1 sigma. In this figure, nitrogen, phosphorus, and silicon include $NO_3$, $PO_4$, and $SIO_4$, respectively.

650

651

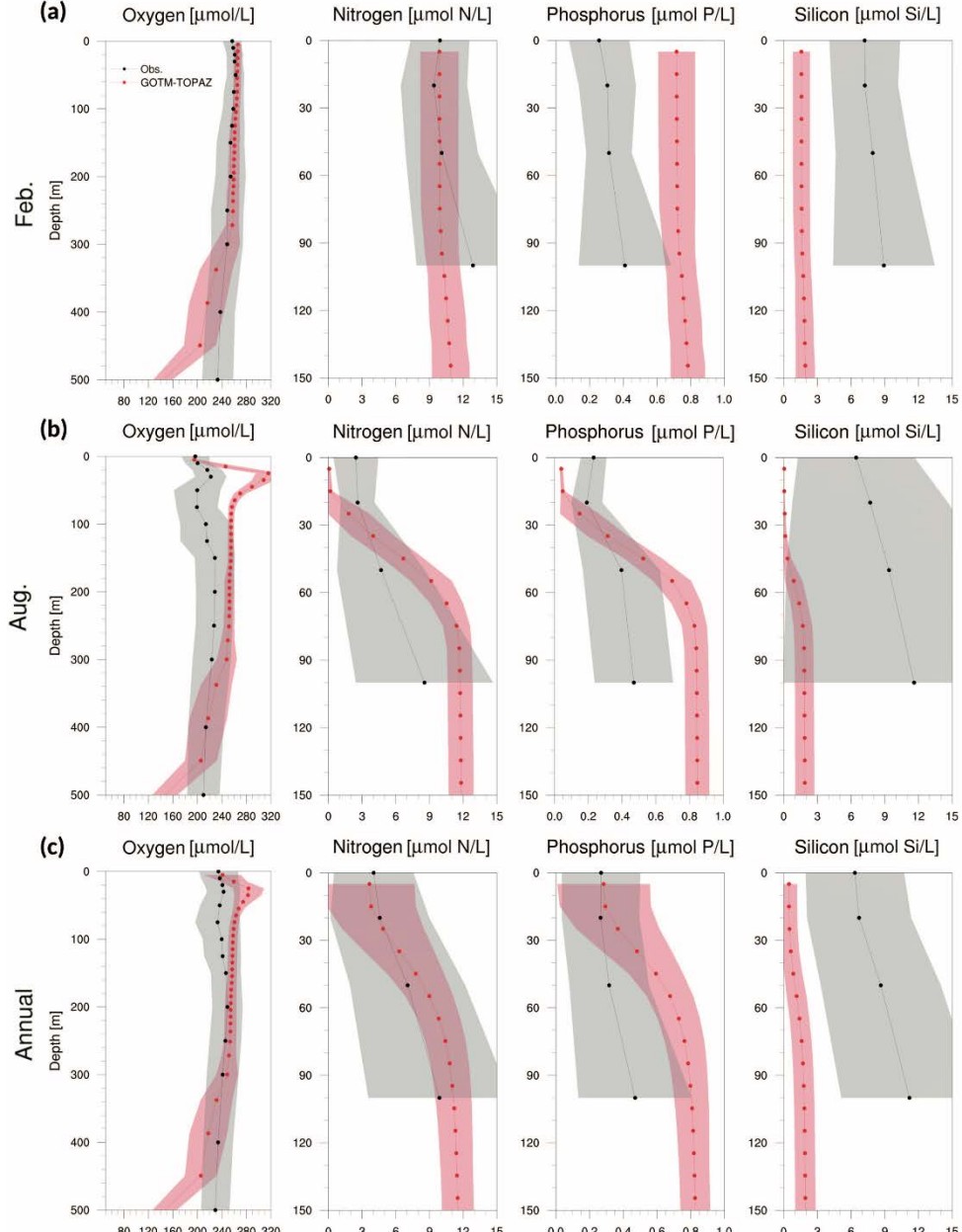

**Figure 10. Vertical profiles from observational data (black dots) and GOTM-TOPAZ results (red dots) at point 102 for concentrations of dissolved oxygen, nitrogen, phosphorus, and silicon averaged from 1999–2008; (a) for February; (b) for August; and (c) annually. The shaded areas represent 1 sigma. In this figure, nitrogen, phosphorus, and silicon include NO₃, PO₄, and SIO₄, respectively.**



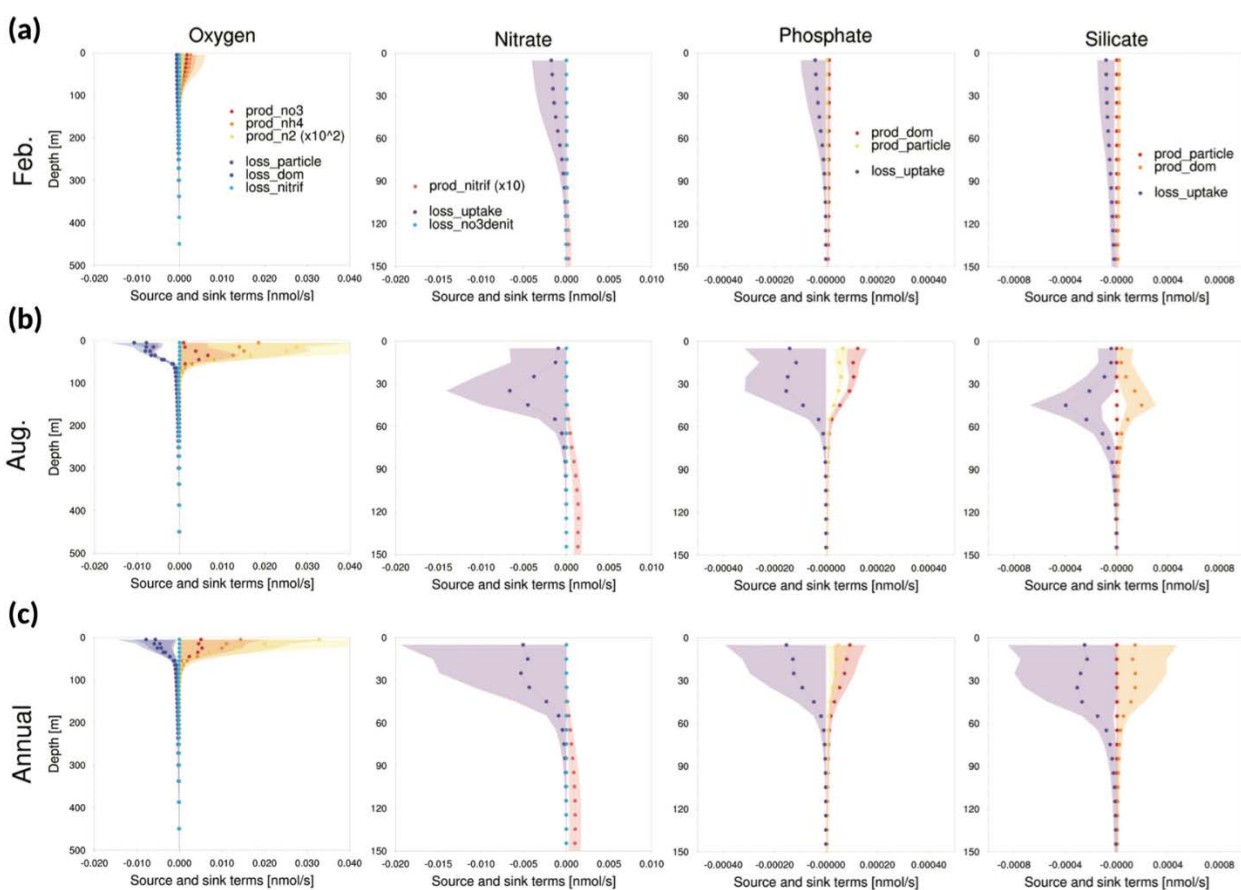


**Figure 11. Vertical profiles of the tendencies of source and sink terms in GOTM-TOPAZ at point 107 for the 10-year period 1999–2008; (a) for February; (b) for August; and (c) annually. The shaded areas represent 1 sigma.**













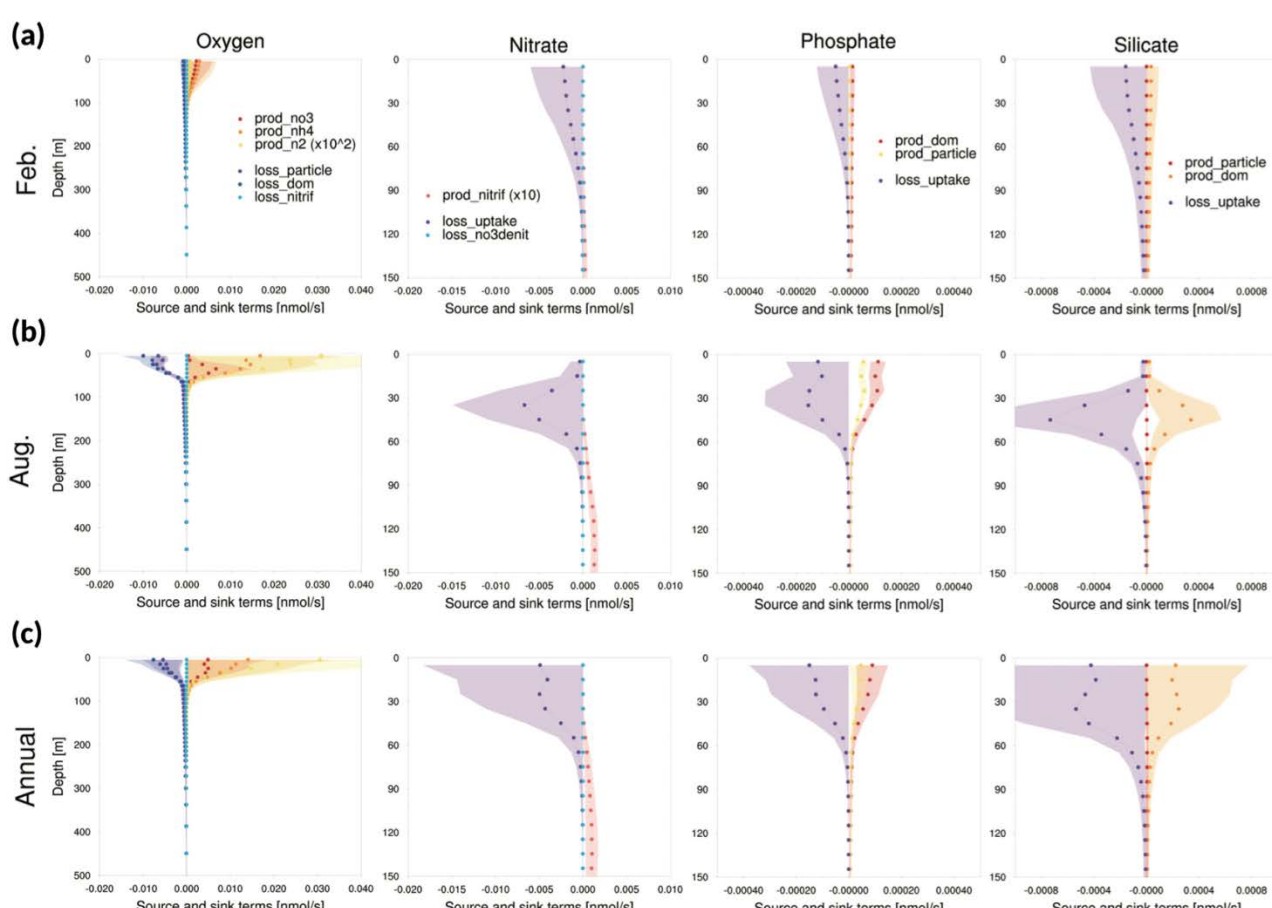


**Figure 12. Vertical profiles of the tendencies of source and sink terms in GOTM-TOPAZ at point 104 for the 10-year period 1999–2008; (a) for February; (b) for August; and (c) annually. The shaded areas represent 1 sigma.**












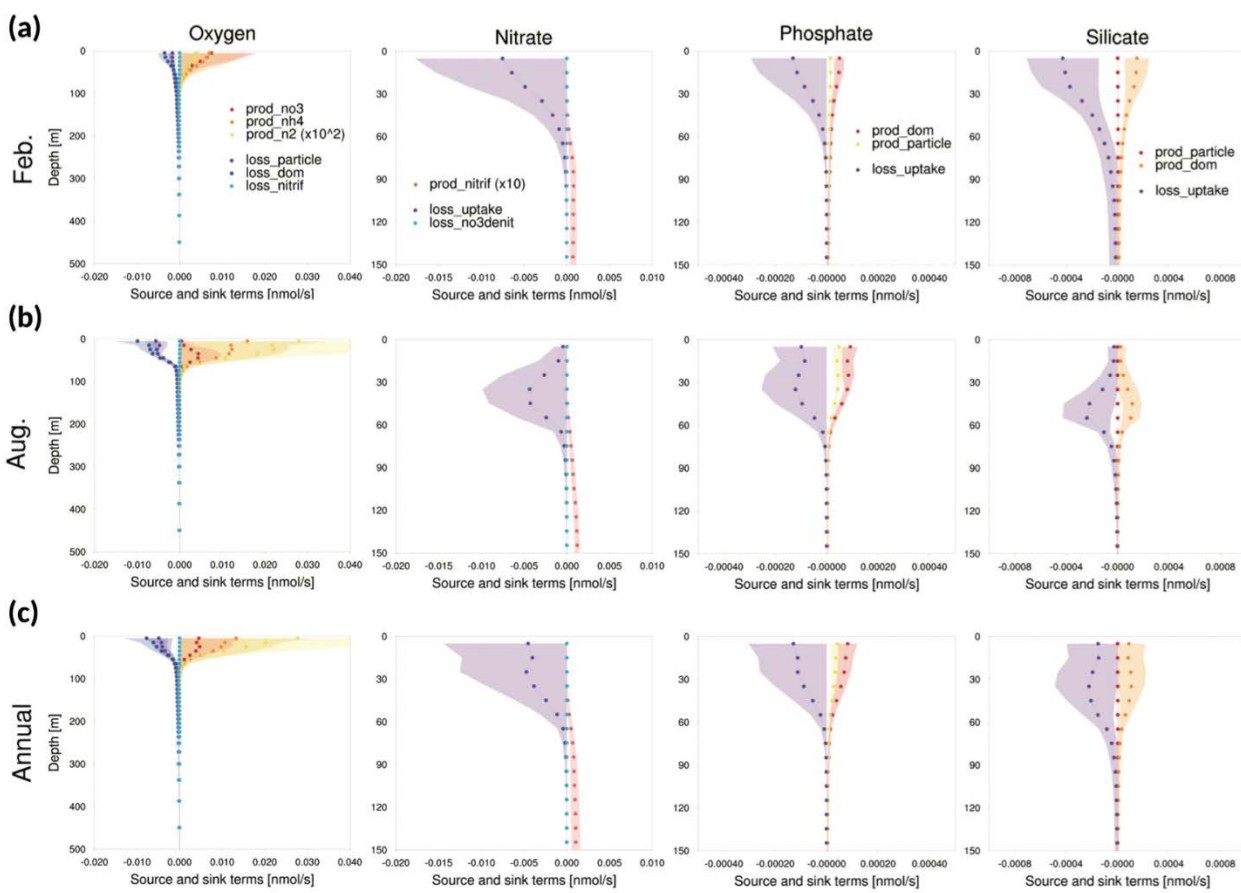


**Figure 13. Vertical profiles of the tendencies of source and sink terms in GOTM-TOPAZ at point 102 for the 10-year period 1999–**
**2008; (a) for February; (b) for August; and (c) annually. The shaded areas represent 1 sigma.**











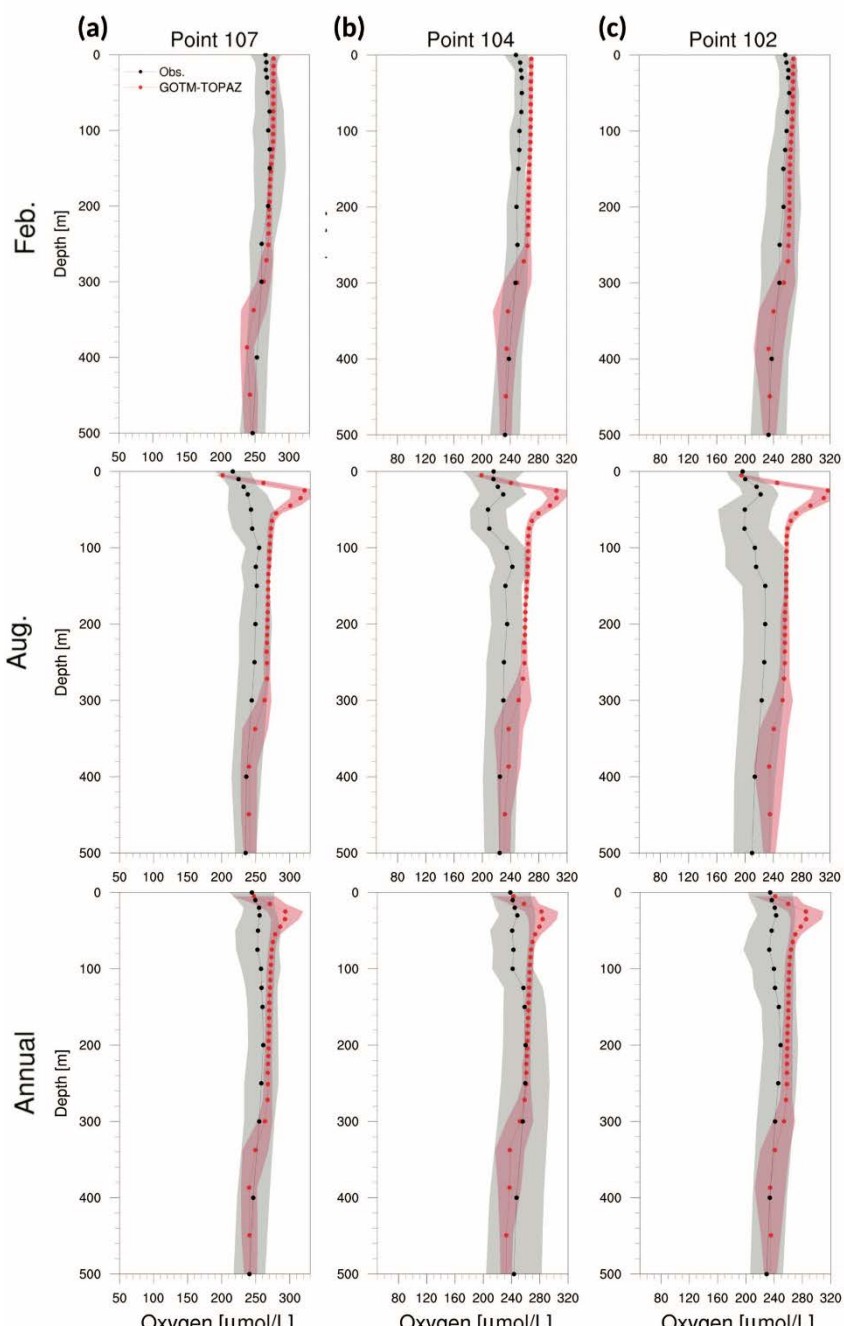


Figure 14. Vertical profiles from observations (black dots) and GOTM-TOPAZ results (red dots) for concentrations of dissolved oxygen averaged from 1999–2008; (a) for point 107; (b) for point 104; and (c) for point 102. GOTM-TOPAZ is simulated by prescribing observations for the initial data. The shaded areas represent 1 sigma.



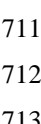

**Figure 15. Scatterplot of mean monthly sea surface CO₂ concentrations as observed by the Surface Ocean CO₂ Atlas and simulated by GOTM-TOPAZ. The thin dotted lines around the 1-to-1 line represent ±1 and 2 μmol kg⁻¹.**