# Peer review of "A single-column ocean-biogeochemistry model (GOTM-TOPAZ) 2 version 1.0"

_Geoscientific Model Development, 2018_

## Referee Comment (RC1) · Anonymous Referee #1 · 19 Oct 2018

Review of A single-column ocean-biogeochemistry model (GOTM-TOPAZ) version 1.0 by Hyun-Chae Jung et al.

Summary.

The manuscript presents a version of the TOPAZ biogeochemistry model coupled to the GOTM water-column hydrodynamics model for the purpose of improving biogeochemical simulations in global climate models. The model is applied to a location in the East Japan Sea, and simulation results are compared with observations. Simulation results are also compared with results from a global 3D implementation of the original TOPAZ model coupled to a global circulation model MOM4.

General comments.

The manuscript is reasonably well-written, and mostly structurally sound, although a large part of the text in Results should be moved to Methods, and information is missing in a number of cases. I have a number of major concerns about the document, however:

1. Innovation. The manuscript only describes the model and a comparison with observations and with a 3D model. None of the broad-brush statements about the potential use and usefulness materialises.

2. Innovation. Other 1D ocean-biogeochemical models already exist. Why create another one? What can this one do that others can't? What can you do with this one that you can't do with others? How does the performance of this one compare with others?

3. Observations. No information about the observational data set is given. How was it collected? Where, and at which time/depth intervals? How was it processed?

4. Validation. The comparison with observations is mostly visual, and involves statements such as 'similar'. This must be made quantitative.

5. Validation. The text often contradicts the information in the figures, and suggests that the results are better than they really are.

6. Validation. Why are only anomalies presented in the time series? Anomalies compared to what? Does this mask biases?

7. Location. The location for application/validation was chosen because of its advective properties at the confluence of two ocean currents (p. 8, l. 8-9). This baffles me, as a water-column model can not (as the authors acknowledge) deal with horizontal advection. How can this site be used to reliably evaluate the model's performance? It's absolutely unsuitable. And indeed, most of the arguments given for mis-matches with the observations are related to advection...

8. Generality. The model is intended to serve very general purposes. However, it

is applied only to a single (unsuitable) site. How can we know that it is generally applicable to the purposes for which it was intended?

Most of these points are addressed further in the detailed points below.

Together, these issues are too many and too severe to repair in a revision cycle. Hence, I have to recommend rejection.

Specific comments.

p. 1, l. 20. reliably: requires quantification.

p. 1, l. 28-29. Requires reference.

p. 2, l. 1. Requires reference.

p. 2, l. 10. for differentiating. I'm not sure what's meant here. Probably not mathematical differentiation?

p. 3, l. 25. Please finish explaining all the variables before going into the equations.

p. 4, l. 25-26. empirical formulas derived from observations. Please expand. Formulas of what kind? Which observations? How were the formulas derived? Are they generally applicable, or (more likely?) specific to the location(s) where the observations were taken? How does this relate to the location used here? There's no need to repeat the Dunne et al. paper, but a summary is required here.

p. 5, l. 11-12. we used: How?

p. 5, l. 17. were determined: how? What was the source of the data?

p. 5, l. 26. process for calculating: please specify.

p. 5, l. 31. monthly average climate values. From which source? How can this be done without systematically enriching the system during the simulation?

p. 6, l. 7. [A]: please provide values and reference(s).

[Figure]

p. 7, l. 2,3: please explain what XI(lambda) and e(lambda) are.

p. 7, l. 1-4: are all these parameter value settings from Manizza et al?

p. 6, eqn 7. Why are contributions to the light-extinction coefficient by CDOM and suspended particulate matter not taken into account? These can be dominant in many locations.

p. 7, section 4.5. How was this used for the test case?

p. 8, l. 2. they: what does this refer to?

p. 8, l. 15. observational: These are not observational data, but model results. You can't verify a model with another model.

p. 8, l. 23. aforementioned observational data. Requires description of the data set.

p. 8, l. 30. similar. Please quantify. There are many occurrences of this kind of terminology, please find and address all.

p. 9, East Sea Intermediate Water. Should have been introduced in the description of the study area.

p. 9, l. 13-15: what do we learn from this?

p. 9, l. 16. Chlorophyll at 40 m. How do we know this is real? This is based solely on results of the current model.

p. 9, l. 24-25. attributed to horizontal advection. How do you know?

p. 9, l. 25-27. I don't see the logic. The 3D model has an influx of nutrients, but the 1D model has higher chlorophyll. How can this influx explain the difference? I would expect the reverse.

Figure 5 b,c. The model appears to be getting enriched with N and P during the simulation. Why? How does this affect the applicability of the model for the intended purposes?

[Figure]

p. 10, l. 1-2. Why February, August and 'the entire period'?

p. 10, l. 3. accurately simulated ... nutrient concentrations. I disagree. The averages of phoshorus and silicon near the surface are outside the standard deviation of the observations.

p. 10, l. 3. upper layer: how is this defined?

p. 10, l. 4-5. phytoplankton at 40m. No observational proof of this is presented.

p. 10, l. 7. each layer: which? how many? Please define all layers clearly.

p. 10, l. 8. properly simulated. I disagree, O2 in the model is substantially higher than observed in the upper 80 m.

p. 10, l. 13. subsurface layer. How defined?

p. 10, l. 13. since. I don't follow the logic here. Were the model results and the observations not processed in the same way?

p. 10, l. 13-14. not in figure 6b between 0 and 80 m.

p. 10, l. 17. excellent. I disagree.

p. 10, l. 14. all within range. No. O2 is outside the standard deviation below 300 m, and silicon over the entire profile.

p. 10, l. 22. reproduced. Well, it doesn't really, does it?

p. 10, l. 26. consistent. I disagree.

p. 10, l. 23. sensitivity experiments. Why were these not done here?

p. 10, l. 30. excellent tool. Please elaborate how.

p. 10, l. 31. parameterisation improvements. How? I don't quite see how this model, which has its own (unexplained, at least here) parameterisations, can be used to improve parameterisations of other models, which may well be incompatible.

p. 11, l. 1. many issues. Please specify. Should these not be sorted out first?

p. 11, l. 5. This: refers to what?

p. 11, l. 11. coupling ... more easily. How/why?

Figure 3: why not include the nutrients and oxygen here? The data from Fig 6 can be plotted in the first column as well; if sparse as coloured circles?

Figure 6: I'm a bit surprised that chlorophyll/fluorescence was not measured as well? If so please use?

Technical corrections.

p. 7, l. 16. anthropogenically. Remove this word.

p. 7, l. 29. Refer to Figure 2 here.

p. 7, l. 23 to p. 8, l. 24. This section is Methods, not Results.

p. 8, section 5.1. header can be removed.

p. 8, l. 27-28. This is Methods, not Results.

Abbreviations. There are so many abbreviations that the manuscript would benefit from a tabulated list.
* * *

---

## Referee Comment (RC2) · Anonymous Referee #2 · 19 Oct 2018

General Comments:

This manuscript is relevant to be published by the Geoscientific Model Development due to the approach of present a single-column ocean biogeochemistry model, GOTM-TOPAZ, as a tool for developing and test new methods to improve the ocean biogeochemistry models. As these models are essentials components in the Earth System Models, the development of tools to improve these models is necessary. Developments and improvements in the ocean biogeochemistry representation by the models are crucial for a better representation of all earth system dynamics.

The work is also interesting to be published because there were modifications in the marine biogeochemical model TOPAZ, as the insertion of a module to reproduce upwelling and also the representation of the air-sea gas transference for O2 and CO2.

The paper is consistent because there was presented an evaluation of the performance of GOTM-TOPAZ by comparisons with observations. Another interesting point of this paper is that, as the model TOPAZ was separated from the MOM model this paper can inspire others studies testing TOPAZ with others OGCM models. Also, others applications with this single-column model could be done in the future.

In summary, I believe that this manuscript is important and deserves to be published. However, I suggest here some points that should be revised aiming to produce a final version in a better condition to be published.

Specific Comments:

Page 3, Line 10: The phrase "we selected points in the East/Japan Sea" is wrong, because in the paper there were just analyzed results for one point. At the page 7 line 22 it is said: "To verify GOTM-TOPAZ, we selected a point . . .".

About this item, I believe it would be necessary to show results for more points. The study would be more robust if there were analyses for more points located in areas with different characteristics. For instance, it would be selected at least more two points to verify the model performance, one would be located in the East Korean Warm Current and other in the North Korea Cold Current. This approach would be more interesting, instead of just to select a point where the two currents meet, as was presented in this paper.

Page 4, Line 3: The section that explains the optical feedback is Section 4.4.

Page 5, Line 2: It is said that the MOM version is 5, however in Figure 4 in the legend it is written that it is analyzed results from MOM4p1_SIS_TOPAZ. Which is the correct MOM version used in this paper?

Page 8, Line 12: It is necessary to describe which are the data used for initializing the biogeochemical tracers in TOPAZ. Which are the data sets and sources?

Page 8, Line 27: Just 4 years of spin up for a biogeochemical model is enough? Most

of the applications with biogeochemical modeling are based in long spin up periods.

Page 8, Line 30: This similarity between GOTM-TOPAZ and observations is just on the first 40 meters for temperature. The difference in deeper regions must be discussed in this point.

Page 9, Line 5: Similarly to the latest comment, it is necessary to be clear in the text that this correspondence in seasonality between the model GOTM_TOPAZ and observation are just in the initial 40 meters.

In Figure 3, there is no figure for observation to chlorophyll. In this case, I do not see a reason for this variable to be included in this figure.

Page 9, Line 13: These correlation coefficients are statically significant?

Page 9, Line 6: It would be interesting here to discuss why the model GOTM-TOPAZ does not represent well the temperature in deeper regions, especially below 80 m. This discussion would be more interesting with the inclusion, in Figure 3, of a figure with the biases between models (MOM and GOTM-TOPAZ) and observations. Maybe this deficiency in the deeper regions is related with a short spin up period.

Page 9, line 30: The phrase "These results can be viewed as validating the gas flux equation reproduced in GOTM-TOPAZ" does not make much sense, once the correlation coefficient for GOTM-TOPAZ was worse than for MOM. Again, the correlation coefficients presented in Figure 5 are statically significant? In this paper, there was no evaluation of the fluxes. It is possible to evaluate the $CO_2$ flux based on observational data, for instance, from SOCAT database.

Page 9, Line 21: The phrase: "In this paper, we have explained the major models that comprises GOTM-TOPAZ and the ocean biogeochemical process reproduced within the models" is not appropriated because you do not have made this on this paper. The model TOPAZ and the ocean biogeochemical process reproduced in this model was not explained on details on this paper. Actually, this explanation was not the main

objective of this paper. To start the item discussion, I believe it would be more relevant to mention the main contributions of this paper, as a study about the development of a single-column ocean-biogeochemistry model.

Page 10, Line 28: In this paper was not presented results about this sensitive experiments that are exemplified, how can you affirm that GOTM-TOPAZ will be good in this kind of applications? In the discussion topics, you should dedicate to discuss based on the results found in the paper.

Finally, in the discussion, there was no evaluation about the upwelling representation. I believe that would be important to include in the paper the evaluation of the upwelling representation. How the module related to w-advection impacted the results? A comparison of the vertical movements reproduced by the model with observations would be interesting.

---

## Author Comment (AC1) · 30 Nov 2018

**Reply to Reviewer's Comments and Suggestions**

Manuscript number: gmd-2018-200
Title: A single-column ocean-biogeochemistry model (GOTM-TOPAZ) version 1.0

We appreciate your considered comments and suggestions, which have proven very helpful in improving our manuscript as well as very valuable in guiding our future research. We have made some revisions to the manuscript in accordance with your comments. The revised portions of the manuscript are marked in red, while our detailed responses below are given in blue.

We greatly appreciate the time and effort you have given to assessing our work and, once again, we thank you very much for your kind comments and suggestions.

**Reviewer #1**

[General comments]

1. "Innovation. The manuscript only describes the model and a comparison with observations and with a 3D model. None of the broad-brush statements about the potential use and usefulness materialises."
: The SCM (1D model) includes important physical processes and has a much lower computation cost than 3D models; therefore, it can be used to perform a variety of experiments repeatedly. Because of this advantage, 1D models can be useful to track mechanisms that are difficult to understand using 3D models. In particular, we think that TOPAZ, which includes complex biogeochemical mechanisms, can be used to obtain insights into the interactions between the chemical makeup of and organisms living in the ocean. In addition, the key processes which are studied via TOPAZ can later be implemented into 3D models. We added content on this to the discussion section.

"The SCM (1D model) includes important physical processes and has a much lower computation cost than 3D models; this means that a variety of experiments can be performed repeatedly. With this advantage, 1D models can be useful to track mechanisms that are difficult to understand using 3D models. We believe that TOPAZ, in particular, can be used to obtain insights on the interactions between the chemical makeup and organisms in the ocean because it accounts for complex biogeochemical mechanisms. In addition, the key processes which are studied via TOPAZ can be implemented later into 3D models."
"For example, the aerosol concentrations are continuously increasing in the over the East Asia region and are known to affect precipitation and atmospheric circulation. Thus, there is a possibility that aerosols affect oceanic biogeochemical processes as deposition occurs into the ocean, and this cannot be ignored. A variety of numerical experiments are necessary to understand this process, but they are difficult to perform using 3D models due to limitations in computing resources. However, as previously noted, GOTM-TOPAZ is fast; as such, it is useful for understanding the biogeochemical changes that occur in the ocean when the concentration of aerosols or $CO_2$ in the atmosphere changes. In addition, recent studies have reported that the distribution of fisheries is changing due to changes in phytoplankton size structure, caused by upwelling intensity on the coast of the East/Japan Sea (Shin et al., 2017). Phytoplanktons of TOPAZ are divided into two-types depending on their size, so it is expected to be useful in above mentioned research."

Shin, J.-W., Park, J., Choi, J.-G., Jo, Y.-H., Kang, J. J., Joo, H. T., and Lee, S. H.: Variability of phytoplankton size structure in response to changes in coastal upwelling intensity in the southwestern East Sea, J. Geophys. Res. Oceans, 122, 10, 262–10, 274, doi:10.1002/2017JC013467, 2017.

2. "Innovation. Other 1D ocean-biogeochemical models already exist. Why create another one? What can this one do that others can't? What can you do with this one that you can't do with others? How does the performance of this one compare with others?"

: You are correct that other 1D ocean biogeochemical models exist; one such model is included in the GOTM that we used (Burchard et al., 2006). However, most of these 1D ocean biogeochemical models include very simple processes and only predict limited oceanic biogeochemical variables. Furthermore, in most cases, they do not take the gas transfer between the atmosphere and ocean into account, or they simply calculate it as a constant. However, TOPAZ distinguishes three kinds of phytoplankton (which are important in ocean biogeochemistry) by size (small and large) and characteristics (diazotrophs), and also accounts for more than thirty biogeochemical variables. It also considers the atmospheric and oceanic environment and calculates the oxygen and carbon dioxide exchange fluxes for each time step. Iron and lithogenic particle deposition and NH4 and NO3 wet/dry deposition from aerosols are described. Since TOPAZ includes this kind of sophistication, we believe that researchers can use it to perform the following various kinds of experiments:

1. Examine changes in the marine environment according to changes in aerosols
2. Examine changes in oceanic biogeochemistry according to changes in the gas flux
3. Examine changes in the phytoplankton size structure according to upwelling

In addition, MOM and other OGCMs only consider temperature changes caused by the absorption of solar radiation by chlorophyll. However, Sonntag and Hense (2011) used a simple 1D ocean biogeochemical model to show that the effect of chlorophyll in a marine environment not only changes water temperature through photosynthesis but also changes the viscosity and albedo of the ocean, which affects the mixed layer depth. In the future, we plan to use GOTM-TOPAZ to perform experiments on changes in viscosity and albedo caused by chlorophyll and also apply this to a 3D model. We added content on this to the discussion section.

"A variety of single-column ocean biogeochemical models have already been developed. However, GOTM-TOPAZ includes complex biogeochemical processes and models over 30 kinds of tracers; the other models, which have only simple structures, do not (Dunne et al., 2012b). Furthermore, GOTM-TOPAZ considers the gas transfer caused by changes in the atmosphere and the physical environment of the ocean, depicting the deposition of dissolved iron, lithogenic aluminosilicate, NH4, and NO3 due to aerosols. We believe that the sophistication of TOPAZ provides researchers with the opportunity to perform a variety of experiments."

Sonntag, S., and Hense, I.: Phytoplankton behavior affects ocean mixed layer dynamics through biological-physical feedback mechanisms, Geophys. Res. Lett., 38, L15610, doi:10.1029/2011GL048205, 2011.

3. "Observations. No information about the observational data set is given. How was it collected? Where, and at which time/depth intervals? How was it processed?"

: The observational data provided by the NIFS (http://www.nifs.go.kr/kodc) (water temperature, salinity, and dissolved oxygen, nitrogen, phosphorus, and silicon concentrations) were measured from vessels at 52, 54, and 69 specific points in the West, South, and East seas. Water temperature, salinity, and dissolved oxygen concentrations were measured every 15 m from 0 m to 500 m, while the nutrient concentrations were measured at intervals of 0, 20, 50, and 100 m. They were measured once every February, April, June, August, October, and December from 1961 to date. The specific measurement dates and times were not fixed and varied according to weather, vessel, and observation device conditions. Therefore, we viewed the measurement data as values that represented each month and used them as such in the model verification. We used the observational data in Fig. 6, the February (winter) and August (summer) mean from 1999 to 2008, and the mean data for the entire period. This information was added to the Experimental Setup section.

"The water temperature and salinity data from the NIFS was measured at 15 m intervals at depths of 0 m to 500 m. They were measured once in February, April, June, August, October, and December every year from 1961 to date."

"these data were measured once every year, in February, April, June, August, October, and December, at depths of 0, 20, 50, and 100 m. Specific measurement dates and times were not fixed, so we viewed the measurement data as values that represented each month and used them to verify the model."

4. "Validation. The comparison with observations is mostly visual, and involves statements such as 'similar'. This must be made quantitative."

: According to your advice, we have revised the text in the Abstract and the Results regarding the evaluation of the model and the observational data to make it quantitative. The text was revised to mention the p-values and correlation coefficients of the model results and the observational data shown in a time series figure. Thank you for the valuable suggestion.

"The temporal variability of observed upper-ocean (0-20m) chlorophyll is well captured by GOTM-TOPAZ model with a correlation coefficient of 0.51."

"The surface correlation coefficients between the GOTM-TOPAZ oxygen, nitrogen, phosphorus, and silicon are 0.47, 0.30, 0.16, and 0.19, respectively."

"The mean chlorophyll concentration at depths of 0–20 m, as simulated by GOTM-TOPAZ and MOM, had similar inter-annual variabilities; their correlation coefficients versus the observational data were 0.53 and 0.60, respectively (Fig. 4a), which is statistically significant ($p < 0.001$)."

"the two models had a correlation coefficient of 0.59 ($p < 0.01$) and a similar inter-annual variability (Fig. 4b)."

"The sea surface dissolved oxygen levels simulated by GOTM-TOPAZ and MOM had correlation coefficients of 0.47 ($p < 0.001$) and 0.50 ($p < 0.001$), respectively, versus the observed data (Fig. 5a)."

"The GOTM-TOPAZ correlation coefficient versus the observed data was 0.31 ($p < 0.001$) for nitrogen, 0.16 ($p < 0.10$) for phosphorus, and 0.19 ($p < 0.05$) for silicon; these were lower than the correlation coefficients between MOM and the observed data (0.36, 0.24, and 0.33, respectively; $p < 0.001$). However, GOTM-TOPAZ seemed to depict the inter-annual variability of nutrients at the sea surface well (Fig. 5b–d)."

5. "Validation. The text often contradicts the information in the figures, and suggests that the results are better than they really are."

: Thank you for pointing this out. After reading your comment, we realized that the text overstates the modeling capabilities of the model in many places, in contrast to what the figures show. We removed or revised the overstated text and made revisions to evaluate the model using objective numerical values. We also revised the parts where this problem was pointed out in specific comments.

"The vertical distributions of salinity are well simulated and are comparable to the observations, although this could also be because relaxation was applied. The water temperature simulated by GOTM-TOPAZ showed a cold bias in the upper layer at a depth of around 120 m. This appears to be the effect of large-scale forcing (from the EKWC) that GOTM-TOPAZ could not resolve. Similar differences in water temperature also appeared at points 104 and 102 (Supplementary Figure 1)."

"GOTM-TOPAZ well simulated dissolved oxygen (surface to 250 m) and nitrogen (surface to 100 m) concentrations during that season (Fig. 6a). However, for phosphorus and silicon at the same depths, there was a difference between the GOTM-TOPAZ results and the observational data."

"During this season, the oxygen concentration simulated by GOTM-TOPAZ, unlike that in

the observational data, increased sharply from depths of 20−60 m. This seems to have been caused by the creation of oxygen from photosynthesis by phytoplankton (Fig. 6b)."

"Nonetheless, the results demonstrated that dissolved oxygen at 80−250 m, nitrogen, and phosphorus are well simulated over 10 years using GOTM-TOPAZ (Fig. 6c)."

6. "Validation. Why are only anomalies presented in the time series? Anomalies compared to what? Does this mask biases?"

:       We used anomalies to check if GOTM-TOPAZ depicted the inter-annual variability of the biogeochemical tracers well. Time series images of the concentrations of chlorophyll, dissolved oxygen, nitrate, phosphorus, and silicon (which are not depicted as anomalies) at the three points are given below for your reference.

[Figure]

**Figure review 1: Chlorophyll time series and correlation values for observational data (black lines), MOM5_SIS_TOPAZ results (blue lines), and GOTM-TOPAZ results (red lines) at point 107 for the 10-year period 1999−2008; (a) the mean value at depth ≥ 20 m and the correlations between the observations and each model; (b) mean values at depths of 20−80 m and the correlation between the two models.**

[Figure]

**Figure review 2: Time series and correlation values from observational data (black lines), MOM5_SIS_TOPAZ results (blue lines), and GOTM-TOPAZ results (red lines) for concentrations of (a) dissolved oxygen, (b) nitrogen, (c) phosphorus, and (d) silicon at point 107 for the 10-year period 1999−2008; in this figure, nitrogen, phosphorus, and silicon include NO3, PO4, and SIO4, respectively.**

[Figure]

**Figure review 3: Same as in Fig. review 1 except for 104 point.**

[Figure]

**Figure review 4: Same as in Fig. review 2 except for 104 point.**

[Figure]

**Figure review 5: Same as in Fig. review 1 except for 102 point.**

[Figure]

**Figure review 6: Same as in Fig. review 2 except for 102 point.**

7. "Location. The location for application/validation was chosen because of its advective properties at the confluence of two ocean currents (p. 8, l. 8-9). This baffles me, as a water-column model can not (as the authors acknowledge) deal with horizontal advection. How can this site be used to reliably evaluate the model's performance? It's absolutely unsuitable. And indeed, most of the arguments given for mis-matches with the observations are related to advection..."

:          Thank you for your observation. The point that we selected was chosen not just because it is a place where two currents with different properties meet, but because this means it has other important oceanographic and biogeochemical significance. This region is being studied with regard to changes in the major fish species and the total catch based on the type of phytoplankton (size and toxicity) (Joo et al., 2014; Shin et al., 2017). Furthermore, when we considered the continuity and quality of the nutrient observation data mentioned in general comment #3, we determined that point 130E/38N was the best sampling site. As GOTM-TOPAZ handles phytoplankton subdivided into small-sized, large-sized, and diazotrophs, we think that it can be used in such studies. Of course, this region does have strong horizontal advection, which is disadvantageous for testing a 1D column model; despite this, we believe that if the analysis results show some degree of significance, they indicate that GOTM-TOPAZ can be used to test a variety of points. Based on your suggestion, we believe that the content in the paper regarding the site selection was inadequate and have thus added related content. In addition, we selected different points in the East/Japan Sea and performed additional tests.

          "We selected three points that have features typical of the East/Japan Sea and for which observation data suitable to use for verification exists (Fig. 2): point 107, where the EKWC and NKCC meet (130.0° E, 38.0° N); point 104, which is an important location along the EKWC (131.3° E, 37.1° N); and point 102, which is in the middle of a warm eddy created as the EKWC moves north (130.6° E, 36.1° N). As noted previously, these points are in regions with strong advection and thus may not be suitable for testing GOTM-TOPAZ, which is an SCM. However, since the results obtained using GOTM-TOPAZ were significant when compared to the observations, we think that this shows that it is possible to perform sensitivity experiments using GOTM-TOPAZ at several kinds of locations."

Joo, H. T., Park, J. W., Son, S. H., Noh, J.–H., Jeong, J.-Y., Kwak, J. H., Saux-Picart, S., Choi, J. H., Kang, C.-K., and Lee, S. H.: Long-term annual primary production in the Ulleung Basin as a biological hot spot in the East/Japan Sea, J. Geophys. Res. Oceans, 119, 3002–3011, doi:10.1002/2014JC009862, 2014.
Shin, J.-W., Park, J., Choi, J.-G., Jo, Y.-H., Kang, J. J., Joo, H. T., and Lee, S. H.: Variability of phytoplankton size structure in response to changes in coastal upwelling intensity in the southwestern East Sea, J. Geophys. Res. Oceans, 122, 10, 262–10, 274, doi:10.1002/2017JC013467, 2017.

8. "Generality. The model is intended to serve very general purposes. However, it is applied only to a single (unsuitable) site. How can we know that it is generally applicable to the purposes for which it was intended?"

:          3D models are generally compared to observations, verified, and used to predict future states. However, 3D models are large,    very complex, and take up significant computing resources. As such, they are difficult to use to understand key earth systems processes. In contrast, single-column models are used mainly in sensitivity tests such as mutual comparisons of model results regarding changes to the parameterization method or to external forcing. Many parts of various ocean biogeochemical processes are depicted empirically, and these equations are fitted mainly to observations on the open sea, such as those from the Pacific or Atlantic oceans. However, the biogeochemical results simulated by current climate models vary depending on the model used, and projects are being conducted to compare and evaluate these results (Orr et al., 2017). We think that the GOTM-TOPAZ model we developed can be used to better understand the characteristics of these global models and improve the parameterization of biogeochemical processes.

Orr, J. C., Najjar, R. G., Aumont, O., Bopp, L., Bullister, J., Danabasoglu, G., Doney, S. C., Dunne, J. P., Dutay, J.-C., Graven, H., Griffies, S. M., Joos, F., Levin, I., Lindsay, K., McKinley, G. A., Oschlies, A., Romanou, A., Schlitzer, R., Tagliabue, A., Tanhua, T., and Yool, A.: Biogeochemical protocols and diagnostics for the CMIP6 Ocean Model Intercomparison Project (OMIP), Geosci. Model. Dev., 10, 2169-2199, doi:10.5194/gmd-10-2169-2017, 2017.

[Specific comments]

9. "p. 1, l. 20. reliably: requires quantification."
: As suggested, the correlation coefficients of the model and the observational data have been added to the abstract.

"The temporal variability of observed upper-ocean (0-20m) chlorophyll is well captured by GOTM-TOPAZ model with a correlation coefficient of 0.51. The surface correlation coefficients between the GOTM-TOPAZ oxygen, nitrogen, phosphorus, and silicon are 0.47, 0.30, 0.16, and 0.19, respectively."

10. "p. 1, l. 28-29. Requires reference."
: We have added these four citations.

Dirmeyer, P. A., Cash, B. A., Kinter III, J. L., Stan, C., Jung, T., Marx, L., Towers, P., Wedi, N., Adams, J. M., Altshuler, E. L., Huang, B., Jin, E. K., and Manganello, J.: Evidence for enhanced land-atmosphere feedback in a warming climate, J. Hydrometeorol., 13, 981-995, doi:10.1175/JHM-D-11-0104.1, 2012.
Friedlingstein, P., Cox, P., Betts, R., Bopp, L., von Bloh, W., Brovkin, V., Cadule, P., Doney, S., Eby, M., Fung, I., Bala, G., John, J., Jones, C., Joos, F., Kato, T., Kawamiya, M., Knorr, W., Lindsay, K., Matthews, H. D., Raddatz, T., Rayner, P., Reick, C., Roeckner, E., Schnitzler, K. G., Schnur, R., Strassmann, K., Weaver, A. J., Yoshikawa, C., and Zeng, N.: Climate-Carbon Cycle Feedback Analysis: Results from the C4MIP Model Intercomparison, J. Clim., 19(14), 3337-3353, doi:10.1175/JCLI3800.1, 2006.
Randerson, J. T., Lindsay, K., Munoz, E., Fu, W., Moore, J. K., Hoffman, F. M., Mahowald, N. M., and Doney, S. C.: Multicentury changes in ocean and land contributions to the climate-carbon feedback, Global Biogeochem. Cycles, 29, 744-759, doi:10.1002/2014GB005079, 2015.
Soden, B. J., and Held, I. M.: An assessment of Climate Feedbakcs in Coupled Ocean-Atmosphere Models, J. Clim., 19(14), 3354, doi:10.1175/JCLI3799.1, 2006.

11. "p. 2, l. 1. Requires reference."
: We have added these four citations.

Dunne, J. P., John, J. G., Adcroft, A. J., Griffies, S. M., Hallberg, R. W., Shevliakova, E. N., Stouffer, R. J., Cooke, W., Dunne, K. A., Harrison, M. J., Krasting, J. P., Malyshev, S. L., Milly, P. C. D., Phillipps, P. J., Sentman, L. A., Samuels, B. L., Spelman, M. J., Winton, M., Wittenberg, A. T., and Zadeh, N.: GFDL's ESM2 global coupled climate-carbon Earth System Models Part I: Physical formulation and baseline simulation characteristics, J. Clim., doi:101175/JCLI-D-11-00560.1, 2012a.
Dunne, J. P., John, J. G., Shevliakova, E., Stouffer, R. J., Krasting, J. P., Malyshev, S. L., Milly, P. C. D, Sentman, L. T., Adcroft, A. J., Cooke, W., Dunne, K. A., Griffies, S. M., Hallberg, R. W., Harrison, M. J., Levy, H., Wittenberg, A. T., Phillips, P. J., and Zadeh, N.: GFDL's ESM2 global coupled climate–carbon earth system models. Part II: carbon system formulation and baseline simulation characteristics, J. Clim., 26, 2247–2267, doi:10.1175/jcli-d-12-00150.1, 2012b.
Jones, C., and Sellar, A.: Development of the 1st version of the UK Earth system model, UKESM newsletter no. 1 − August 2015, available at: https://ukesm.ac.uk/ukesm-newsletter-no-1-august-2015/ (last access: 4 November 2018), 2015.
Sokolov, A., Kicklighter, D., Schlosser, C. A., Wang, C., Monier, E., Brown-Steiner, B., Prinn, R., Forest, C., Gao, X., Libardoni, A., and Eastham, S.: Description and Evaluation of the MIT Earth System Model (MESH), J. Adv. Model. Earth. Sy., 10(8), 1759-1789, doi:10.1029/2018MS001277, 2018.

12. "p. 2, l. 10. for differentiating. I'm not sure what's meant here. Probably not mathematical differentiation?"

:       You are correct. We have revised the text to convey our intended meaning more precisely. The revised text is shown below. Thank you for the useful advice.

        "There are still no accurate methodologies with which to distinguish biogeochemical variables and to represent biogeochemical processes as formulas (Sauerland et al., 2018)"

13. "p. 3, l. 25. Please finish explaining all the variables before going into the equations."

:       We have added descriptions of all variables included in Eqs. (1)−(4). We revised the text as shown below.

        "In Eqs. (1) and (2), $u$, $v$ and $w$ represent the mean velocities in the spatial directions $x$ (eastward), $y$ (northward), and $z$ (upward), respectively; $v$ represents the molecular diffusivity of momentum; $\rho_0$ represents a constant reference density; $p$ represents pressure; and $f$ represents the Coriolis parameter. In Eq. (3), the temperature (T) equation, $v'$ represents the molecular diffusivity due to heat; $c_p$ represents the heat capacity; and $I$ represents the vertical divergence of short-wave radiation. The effect of solar radiation absorbed by seawater is included in this equation; thus, Eq. (3) is closely associated with the radiation parameterization method. Moreover, a coupled ocean biogeochemistry model must contain an additional short-wave absorption process associated with chlorophyll synthesis distributed throughout the upper-ocean layer (Morel and Antoine, 1994; Cloern et al., 1995; Manizza et al., 2005; Litchman et al., 2015; Hense et al., 2017). Based on the methodology of Manizza et al. (2005), we applied a visible light absorption process due to chlorophyll synthesis, explained in detail in Sect 4.4, to the coupled model. Equation (4) explains the vertical distribution of salinity (S). In this equation, $v''$ represents the molecular diffusivity of salinity; $\tau_R$ represents the relaxation time scale; and $S_R$ represents the observed salinity distribution. In other words, the terms on the right side of this equation express the "relaxation" process based on observations."

14. "p. 4, l. 25-26. empirical formulas derived from observations. Please expand. Formulas of what kind? Which observations? How were the formulas derived? Are they generally applicable, or (more likely?) specific to the location(s) where the observations were taken? How does this relate to the location used here? There's no need to repeat the Dunne et al. paper, but a summary is required here."

:       As per your advice, we added more information on the main processes of the ocean biogeochemical process implemented by TOPAZ and added a reference (de Baar, 1994; Redfield et al., 1963). The Redfield ratio (C:N:P found in phytoplankton is 106:16:1), Leibig's law of the minimum (which states that growth is controlled by the limiting nutrient), and size considerations (large organisms feed on small ones) were mentioned. The revised text is shown below. Thank you for the useful advice.

        "The biological processes of TOPAZ were reproduced with a focus on phytoplankton growth, nutrient and light limitations, the grazing process, and empirical formulas derived from observations. These are followed by the Redfield ratio (Redfield et al., 1963), Liebig's law of the minimum (de Baar, 1994), and size considerations (large organisms feed on smaller ones), which were used to establish the ocean ecosystem model (Dunne et al., 2012b)."

15. "p. 5, l. 11-12. we used: How?"
:        The "SUBROUTINE adv center," which calculates tendencies caused by w-advection and applies them to tracers in GOTM, was applied to TOPAZ. The "SUBROUTINE topaz_w_adv" module, in particular, was created within TOPAZ and the "SUBROUTINE adv_center" was linked to it. The text was revised to demonstrate this better, and the revised text is shown below. Thank you for the useful comment.

        "We connected the w-advection module in GOTM to TOPAZ so that the upwelling was reproduced in TOPAZ."

16. "p. 5, l. 17. were determined: how? What was the source of the data?"
:        Thank you for making this suggestion. We have revised the text as follows to convey our intended meaning. We also added text to the Experimental Setup section to describe the source of the initial GOTM data.

        "The observed data such as seawater temperature and salinity was used to initialize and relax vertical structures in the GOTM throughout the simulation. This data was provided by the National Institute of Fisheries Science (NIFS; http://www.nifs.go.kr/kodc)."

17. "p. 5, l. 26. process for calculating: please specify."
:        TOPAZ includes calculations for sediment calcite cycling and the external bottom flux of O2, NH4, PO4, and alkalinity. We revised the relevant text as shown below.

        "TOPAZ includes processes for variables include sediment calcite cycling and the external bottom fluxes of O2, NH4, PO4, and alkalinity (Dunne et al., 2012b)."

18. "p. 5, l. 31. monthly average climate values. From which source? How can this be done without systematically enriching the system during the simulation?"
:        We used MOM's initial data, which was provided by the Australian Research Council's Centre of Excellence for Climate System Science (ARCCSS; http://climate-cms.unsw.wikispaces.net/Data). The text added to the manuscript is shown below. TOPAZ's internal mechanism is shown in Fig. review 7. The material provided from the atmosphere is consumed by phytoplankton, decomposes, and sinks; as the material cycles through this process, it does not seem to accumulate.

        "These surface flux data are provided by the Australian Research Council's Centre of Excellence for Climate System Science (ARCCSS; http://climate-cms.unsw.wikispaces.net/Data)."

[Figure]

**Figure review 7. Ocean biogeochemical process represented in TOPAZ.**

19. "p. 6, l. 7. [A]: please provide values and reference(s)."
:       In eq 6, [A] is the surface ocean concentration of gas A as predicted by the model. Concentrations are indicated throughout with [] and are in units of µmol of the chemical species per kg of seawater. The information related to air-sea gas transfer is referenced in Najjar and Orr (1998). We revised the text which describes Eq. 6 as shown below.

"Here, F is the upward flux of gas A and $k_w$ is its gas transfer velocity, which can be calculated as a function of the Schmidt number and wind speed at 10 m (Wanninkhof, 1992). $\rho$ is the density of surface seawater, [A] is the concentration [µmol kg$^{-1}$] of gas A at the surface of the ocean, and [A]$_{sat}$ is the corresponding saturation concentration of gas A in equilibrium with a water vapor-saturated atmosphere at total atmospheric pressure (Najjar and Orr, 1998). [A] is predicted by the model. Please see Najjar and Orr (1998) for further detailed information related to Eq. (6)."

20. "p. 7, l. 2,3: please explain what XI(lambda) and e(lambda) are."
:       $\chi_{(\lambda)}$ represents the pigment absorption and $e_{(\lambda)}$ represents the power law for absorption. We have modified the sentence as follows.

"In these bands, the values of the pigment adsorption $\chi_{(\lambda)}$ are 0.037 and 0.074 m$^{-2}$ mg Chl m$^{-3}$, respectively; $e_{(\lambda)}$, the power law for absorption, has values of 0.629 and 0.674 [no units], respectively."

21. "p. 7, l. 1-4: are all these parameter value settings from Manizza et al?"
:       Yes, we used all parameter settings from Manizza et al. (2005).

22. "p. 6, eqn 7. Why are contributions to the light-extinction coefficient by CDOM and suspended particulate matter not taken into account? These can be dominant in many locations."

: As you have noted, light extinction due to CDOM and suspended particulate matter are certain items that must be considered in the model. However, as far as we know, there is currently no earth systems model in existence that considers all of these mechanisms. We can use GOTM-TOPAZ to perform experiments that consider changes in light-extinction due to chlorophyll, CDOM, and suspended particulate matter. We can perform a study that first uses a 1D column model to test the stability of this kind of parameterization, and then apply it to a 3D model. We are grateful to the you for your providing helpful suggestions regarding the improvement of our model.

23. "p. 7, section 4.5. How was this used for the test case?"

: Thank you for drawing our attention to this. We added tests for new points in the East/Japan Sea and included the results in the supplementary material.

In the experiment at point 102, we prescribed the upwelling as decreasing linearly in the upward and downward directions at maximum value of 0.0000005 m/s, based on a depth of 100 m. The water temperature from GOTM-TOPAZ shown in Fig. review 8 demonstrates that a cold region exists due to upwelling at a depth of around 200 m. However, this cold region, which actually exists at point 102, is due to cold advection, and its mechanism is different from the upwelling experiment. Nonetheless, we performed experiments to verify the implementation of upwelling in the model. The mean chlorophyll concentration at depths of 20−80 m show that there is a rapid increase in upwelling during winter (Fig. review 9). Because of the effect of this upwelling, the nutrients below a depth of 200 m were supplied to the upper layer during the previous period. The supplied nutrients are consumed and thus have an effect on the increase in chlorophyll concentration around at 20−80 m in the winter (Fig. review 10). The effect of the upwelling is also seen in the vertical profile of dissolved oxygen. Fig. review 10 shows that the middle layer of seawater, which is deeper than 300 m (where there is little dissolved oxygen), was supplied to the upper layer, and the concentration of dissolved oxygen below a depth of 150 m decreased sharply. We still do not have adequate data to implement upwelling that is similar to reality. In the future, we plan to collect observational data related to this and perform a study on upwelling on the eastern coast and changes in the ocean biogeochemical environment.

[Figure]

**Figure review 8: Comparison of the vertical distribution over time for water temperature [℃] (a), salinity [psu] (b) and the difference between the two (GOTM-TOPAZ minus obs.) at point 102 for the 10-year period from 1999–2008. The upwelling is prescribed to the GOTM-TOPAZ.**

[Figure]

**Figure review 9: Chlorophyll time series from GOTM-TOPAZ (red lines) and upwelling case (blue lines) at point 102 for the 10-year period from 1999–2008. Chlorophyll concentration is averaged between 20 to 80m depth. The upwelling is prescribed to the GOTM-TOPAZ.**

[Figure]

**Figure review 10: Vertical profile from the GOTM-TOPAZ (red dots) and upwelling case (blue dots) with respect to dissolved oxygen, nitrogen, phosphorus, and silicon averaged from 1999 to 2008; (a) for February; (b) for August; and (c) annually. The shaded area represents 1 sigma. In this figure, nitrogen, phosphorus, and silicon include NO3, PO4, and SIO4, respectively.**

24. "p. 8, l. 2. they: what does this refer to?"
:        In this text "they" refers to the East/Japan Sea. We revised the text as follows to convey our intended meaning more precisely. Thank you for this comment.

        "The East/Japan Sea is divided into warm and cold regions relative to the 40° N parallel, and, since the current pattern and characteristics of the East/Japan Sea vary spatially and seasonally, this region is very important to oceanographic studies."

25. "p. 8, l. 23. aforementioned observational data. Requires description of the data set."
:        Thank you for this suggestion. As mentioned in the answer to general comment #3, we have added detailed information on the water temperature, salinity, dissolved oxygen, and nutrient observation data provided by the NIFS to the paper. Please refer to our answer to general comment #3.

26. "p. 8, l. 30. similar. Please quantify. There are many occurrences of this kind of terminology, please find and address all."
:        Thank you for this suggestion. We revised most of the sentences in the paper that evaluated the model so that they included quantitative values (e.g., coefficients of correlation) rather than the term "similar." Please refer to our answer to general comment #4.

27. "p. 9, East Sea Intermediate Water. Should have been introduced in the description of the study area."
:        Based on your suggestion, we moved the text that introduces the East Sea Intermediate Water to the Experimental Setup section.

28. "p. 9, l. 13-15: what do we learn from this?"
:        MOM is a low-resolution model with a grid size of about 1° by 1°. Therefore, because TOPAZ is connected to MOM, the atmospheric forcing or ocean physical environment is transferred as a mean value of the grid. Thus, the biogeochemical results also generate smoothed results. However, GOTM-TOPAZ uses detailed data from a single point as input and can therefore show extreme values well.

29. "p. 9, l. 16. Chlorophyll at 40 m. How do we know this is real? This is based solely on results of the current model."
:        The fact that chlorophyll is mainly distributed around 40 m at the point in the East/Japan Sea sampled in this study is a result generated by the model. However, because there is no observational data on this, it is difficult to confirm whether it is true or not. Therefore, we removed the chlorophyll figure in Fig. 3 and to add a reference. The revised Fig. 3 is shown below. Thank you for your suggestion.

        "Phytoplankton in the East/Japan Sea are generally present in the highest concentrations at depths of around 10−60 m (Rho et al., 2012)."

[Figure]

**Figure 3: Comparison of the vertical distribution over time for water temperature [℃], salinity [psu] and the difference between the two (GOTM-TOPAZ results minus observational data) at point 107 for the 10-year period from 1999−2008.**

30. "p. 9, l. 24-25. attributed to horizontal advection. How do you know?"
:        As you noted, the analysis of the difference in chlorophyll concentrations in the paper, as simulated by the two models, is excessive. Not only are the transport tendencies of the MOM-TOPAZ and GOTM-TOPAZ different, but so are the atmospheric forcing data described by these models. They also model the ocean physical environment in different ways; therefore, the reason that the results from the two models are different is complex. We deleted the text on the analysis of direct reasons. Thank you for the useful suggestion.

        "In the TOPAZ module in MOM, the transport tendencies of each tracer were calculated in the ocean model; however, this process was not carried out in GOTM-TOPAZ. In addition, MOM and GOTM-TOPAZ are not only just different models of the marine physical environment; the atmospheric forcing data they each use are also different. Therefore, there are complex reasons for the differences in the results of the two models, and further detailed experiments and analysis are required."

31. "p. 9, l. 25-27. I don't see the logic. The 3D model has an influx of nutrients, but the 1D model has higher chlorophyll. How can this influx explain the difference? I would expect the reverse."
:        Thank you for making a good point. As mentioned in the answer to specific comment #30, the reasons for the difference in chlorophyll simulated by the 3D model (MOM) and the 1D model (GOTM-TOPAZ) are not limited to one or two items but are complex. We determined that the text that you referred to could be a problem and have deleted it from the paper. Regarding the reason for the higher chlorophyll in the 1D model compared to the 3D model, please refer to the answer to specific comment #28.

32. "Figure 5 b,c. The model appears to be getting enriched with N and P during the simulation. Why? How does this affect the applicability of the model for the intended purposes?"
:        Currently, we assume that this problem occurs due to differences in the amounts of nutrients used during the non-implemented advection and sinking processes, and this is considered a limitation of the single-column model. Furthermore, in Fig. 5, this increasing trend occurs in the latter five years of the modeling period. This problem must be considered during any long-term experiments using this model.

33. "p. 10, l. 1-2. Why February, August and 'the entire period'?"
:        As mentioned in general comment #3, the NIFS conducts observations once every February, April, June, August, October, and December. Therefore, February was chosen to represent winter and August was chosen to represent summer. The model results for the entire period were verified to confirm its reliability.

34. "p. 10, l. 3. accurately simulated ... nutrient concentrations. I disagree. The averages of phoshorus and silicon near the surface are outside the standard deviation of the observations."
:        Thank you for the useful comment. As mentioned in the response to general comment #5, we have revised the parts in which the performance of the model was overstated to make them objective.

        "GOTM-TOPAZ well simulated dissolved oxygen (surface to 250 m) and nitrogen (surface to 100 m) concentrations during that season (Fig. 6a). However, for phosphorus and silicon at the same depths, there was a difference between the GOTM-TOPAZ results and the observational data."

35. "p. 10, l. 3. upper layer: how is this defined?"
:        We have revised "upper layer" to "surface to 100 m."

36. "p. 10, l. 4-5. phytoplankton at 40m. No observational proof of this is presented."
:        We deleted the chlorophyll figure from Fig. 3 and also deleted the text that referred to it. Thank you for this suggestion.

37. "p. 10, l. 7. each layer: which? how many? Please define all layers clearly."
:        Based on your suggestion, we revised the text to clearly present the information on the layers , and the amounts reduced, as shown below.

        "The concentrations of nitrogen, phosphorus, and silicon simulated by GOTM-TOPAZ from the surface to 60 m decreased during August, and these concentrations were clearly distinguishable from the surface to 60 m due to strong stratification in the summer (Fig. 6b)."

38. "p. 10, l. 8. properly simulated. I disagree, O2 in the model is substantially higher than observed in the upper 80 m."
:        We agree. We have revised the text as shown below.

        "During this season, the oxygen concentration simulated by GOTM-TOPAZ, unlike that in the observational data, increased sharply from depths of 20−60 m. This seems to have been caused by the creation of oxygen from photosynthesis by phytoplankton (Fig. 6b)."

39. "p. 10, l. 13. subsurface layer. How defined?"
:        We have revised the text according to specific comment #40, and the relevant text was deleted. Please refer to the answer to specific comment #40 for more details.

40. "p. 10, l. 13. since. I don't follow the logic here. Were the model results and the observations not processed in the same way?"
:        Thank you for the useful comment. We have revised the text. Please refer to the appropriate section of the revised paper.

"However, a highly concentrated dissolved oxygen concentration is not apparent in the observational data, because the low dissolved oxygen is transported by the EKWC (Rho et al., 2012)."

41. "p. 10, l. 13-14. not in figure 6b between 0 and 80 m."
: You are correct; thank you for your careful observation. The interval in which the dissolved oxygen in GOTM-TOPAZ and the observational data was similar was revised to 80–250 m. The revised text is as follows.

"The concentrations of dissolved oxygen from 80–250 m were similar in both the results from GOTM-TOPAZ and in the 10-year observational data (Fig. 6c)."

42. "p. 10, l. 17. excellent. I disagree."
: We revised the sentence that includes the term "excellent," as shown below. Thank you for the suggestion.

"Nonetheless, the results demonstrated that dissolved oxygen at 80–250 m, nigrogen, and phosphorus are well simulated over 10 years using GOTM-TOPAZ (Fig. 6c)."

43. "p. 10, l. 14. all within range. No. O2 is outside the standard deviation below 300 m, and silicon over the entire profile."
: Thank you for this observation. We revised the parts that overstate the level of modeling. Please refer further to the response to specific comment #42.

44. "p. 10, l. 22. reproduced. Well, it doesn't really, does it?"
: You are correct. We have revised the term "ocean biogeochemical processes" to "biological-physical feedback" in the text.

"In this paper, we explain the major models that comprise GOTM-TOPAZ and the biological-physical feedback loop that they reproduce."

45. "p. 10, l. 26. consistent. I disagree."
: We have deleted this text.

46. "p. 10, l. 23. sensitivity experiments. Why were these not done here?"
: We invested a period of over one year to the process of separating TOPAZ from MOM and combining it with GOTM. As such, we needed to present the results obtained from these processes before performing experiments using the model, and we submitted a development and technical paper on this to GMD. We plan to perform sensitivity experiments using the model soon. Thank you for the suggestion.

47. "p. 10, l. 30. excellent tool. Please elaborate how."
: Thank you for your suggestion. We did not perform sensitivity experiments related to this text in this paper. Therefore, we deleted this text and have referred to research fields in which GOTM-TOPAZ can be used in the Discussion section. Please also refer to the answers to general comments

**1 and #2 for more.**

48. "p. 10, l. 31. parameterisation improvements. How? I don't quite see how this model, which has its own (unexplained, at least here) parameterisations, can be used to improve parameterisations of other models, which may well be incompatible."
:         As mentioned in the response to specific comment #47, we have not performed an experiment which can prove the content for this text. Therefore, the text was deleted and a description of the fields in which the model can be used was added to the paper. Please refer to the answers to general comments #1 and #2 for more.

49. "p. 11, l. 1. many issues. Please specify. Should these not be sorted out first?"
:         We deleted this text and instead clearly described the research fields in which GOTM-TOPAZ can be used. Please refer to the answers to general comments #1 and #2 for more.

50. "p. 11, l. 5. This: refers to what?"
:         In this section, "This study" refers to "Sonntag and Hense (2011)." Following your advice, we have revised the text as shown below to convey our meaning clearly. Thank you.

        "Sonntag and Hense (2011) provided us a better understanding of the needs and direction to focus on with GOTM-TOPAZ, and we plan to apply various climate-ocean biogeochemistry feedback mechanisms to it in future research."

51. "p. 11, l. 11. coupling ... more easily. How/why?"
:         We separated TOPAZ from MOM and divided it into modules that manage initialization, optical feedback, and column physics to create a stand-alone version of TOPAZ. Of course, this version was created in the form of a static library. Other researchers can try combining this library with a variety of ocean models. As mentioned in the Introduction, ocean biogeochemical processes in current earth systems models have a large bias and inter-model diversity based on the type of model. As such, we believe that the separated TOPAZ modules can not only be used to develop new earth systems models, but also that it can help reduce the uncertainty in current models via a comparative analysis of various model results.

52. "Figure 3: why not include the nutrients and oxygen here? The data from Fig 6 can be plotted in the first column as well; if sparse as coloured circles?"
:         As mentioned in the response to general comment #3, the observational data on oxygen and nutrients provided by the NIFS are measured at depths of 0, 20, 50, and 100 m six times a year (February, April, June, August, October, and December). The continuity of this data is not good and missing values exist. Therefore, to reduce the limitations of the observational data, we decided that it would be better to take their averages when performing a comparison, as in Fig. 6, rather than to use the image format in Fig. 3. Thank you for your suggestion.

53. "Figure 6: I'm a bit surprised that chlorophyll/fluorescence was not measured as well? If so please use?"
:         The NIFS data that we used did not include chlorophyll/fluorescence measurements.

[Technical corrections]

54. "p. 7, l. 16. anthropogenically. Remove this word."
:        This has been done.

55. "p. 7, l. 29. Refer to Figure 2 here."
:        This has been done.

56. "p. 7, l. 23 to p. 8, l. 24. This section is Methods, not Results."
:        Thank you for this important suggestion. We have moved this text to the section "5. Experimental Setup".

57. "p. 8, section 5.1. header can be removed."
:        Thank you for the suggestion. As you suggested, we deleted the header for section 5.1.

58. "p. 8, l. 27-28. This is Methods, not Results."
:        As you suggested, we moved this content to the Experimental Setup section.

59. "Abbreviations. There are so many abbreviations that the manuscript would benefit from a tabulated list."

:        We have added the following table, which shows the abbreviations used in the paper, to our manuscript. Thank you for the valuable suggestion.

**Table 1: List of abbreviations**

| Abbreviation | Full form |
| --- | --- |
| ESM | Earth System Model |
| SCM | Single Column Model |
| OGCM | Ocean Global Circulation Models |
| CMIP5 | Coupled Model Intercomparison Project 5 (fifth phase) |
| GFDL | Geophysical Fluid Dynamics Laboratory |
| ARCCSS | Australian Research Council Centre of Excellence for Climate System Science |
| NIFS | National Institute of Fisheries Science |
| ESM2M | Earth System Model version 2, with Modular Ocean Model Version 4.1 |
| ESM2G | Earth System Model version 2, with General Ocean Layer Dynamics |
| ECMWF | European Centre for Medium-Range Weather Forecasts |
| GOTM | General Ocean Turbulence Model |
| TOPAZ | Tracers of Phytoplankton with Allometric Zooplankton |
| MOM5 | Modular Ocean Model version 5 |
| NEMO | Nucleus for European Modelling of the Ocean |
| MEDUSA | Model of Ecosystem Dynamics, Nutrients Utilization, Sequestration and Acidification |
| PISCES | Pelagic Interactions Scheme for Carbon and Ecosystem Studies |
| SOCAT | Surface Ocean $CO_2$ Atlas |
| SeaWiFS | Sea-viewing Wide Field-of-view Sensor |
| CORE-II | Coordinated Ocean-ice Reference Experiments II |
| PAR | Photosynthetically Active Radiation |
| TWC | Tsushima Warm Current |
| EKWC | East Korea Warm Current |
| NKCC | North Korea Cold Current |
| NB | Nearshore Branch |
| OB | Offshore Branch |
| ESIW | East Sea Intermediate Water |

---

## Author Comment (AC2) · 30 Nov 2018

**Reply to Reviewer's Comments and Suggestions**

Manuscript number: gmd-2018-200
Title: A single-column ocean-biogeochemistry model (GOTM-TOPAZ) version 1.0

We appreciate your considered comments and suggestions, which have proven very helpful in improving our manuscript as well as very valuable in guiding our future research. We have made some revisions to the manuscript in accordance with your comments. The revised portions of the manuscript are marked in red, while our detailed responses below are given in blue.

We greatly appreciate the time and effort you have given to assessing our work and, once again, we thank you very much for your kind comments and suggestions.

**Reviewer #2**

[General Comments]

This manuscript is relevant to be published by the Geoscientific Model Development due to the approach of present a single-column ocean biogeochemistry model, GOTMTOPAZ,as a tool for developing and test new methods to improve the ocean biogeochemistry models. As these models are essentials components in the Earth System Models, the development of tools to improve these models is necessary. Developments and improvements in the ocean biogeochemistry representation by the models are crucial for a better representation of all earth system dynamics. The work is also interesting to be published because there were modifications in the marine biogeochemical model TOPAZ, as the insertion of a module to reproduce upwelling and also the representation of the air-sea gas transference for O2 and CO2.

The paper is consistent because there was presented an evaluation of the performance of GOTM-TOPAZ by comparisons with observations. Another interesting point of this paper is that, as the model TOPAZ was separated from the MOM model this paper can inspire others studies testing TOPAZ with others OGCM models. Also, others applications with this single-column model could be done in the future.

In summary, I believe that this manuscript is important and deserves to be published. However, I suggest here some points that should be revised aiming to produce a final version in a better condition to be published.

: Thank you for positive review and helpful comments. We addressed each of specific comments and separately responded in next page.

[Specific Comments]

1. "Page 3, Line 10: The phrase "we selected points in the East/Japan Sea" is wrong, because in the paper there were just analyzed results for one point. At the page 7 line 22 it is said: "To verify GOTM-TOPAZ, we selected a point : : :"."
:          Thank you for your observation. Following the suggestion in the comments below, we have added two points to the revised manuscript.

2. "About this item, I believe it would be necessary to show results for more points. The study would be more robust if there were analyses for more points located in areas with different characteristics. For instance, it would be selected at least more two points to verify the model performance, one would be located in the East Korean Warm Current and other in the North Korea Cold Current. This approach would be more interesting, instead of just to select a point where the two currents meet, as was presented in this paper."
:          Thank you for your valuable suggestion. Based on it, we ran our model on points 104 (131.3E, 37.1N) and 102 (103.6E, 36.1N), as well as at point 107 (130.0E, 38.2N), to verify it. Points at which the NKCC flows are located in North Korea and available observation values do not exist. Therefore, we selected a point where the EKWC is dominant (point 102) and a point in the middle of the warm eddy caused by the EKWC (point 104). The selections were centered on points for which nutrient observation values exist, and which have enough continuity to be used for verification. The images resulting from the analyses of the two new points are not very different from that from point 107. Therefore, they were added to the supplementary material and not to the main manuscript. Please refer to the revised paper for more.

3. "Page 4, Line 3: The section that explains the optical feedback is Section 4.4."
:          Done, accordingly.

4. "Page 5, Line 2: It is said that the MOM version is 5, however in Figure 4 in the legend it is written that it is analyzed results from MOM4p1_SIS_TOPAZ. Which is the correct MOM version used in this paper?"
:          We used MOM version 5 in our study. A revised Fig. 4 is attached below. Thank you for your observation.

[Figure]

**Figure 4: Chlorophyll anomaly time series and correlation values for observational data (black lines), MOM5_SIS_TOPAZ resuts (blue lines), and GOTM-TOPAZ resuts (red lines) at point 107 for the 10-year period 1999−2008; (a) the mean value atdepths ≥ 20 m and the correlations between the observations and each model; (b) mean values at depths of 20− 80 m and the correlation between the two models.**

5. "Page 8, Line 12: It is necessary to describe which are the data used for initializing the biogeochemical tracers in TOPAZ. Which are the data sets and sources?

: We used the tracer input data provided by the Australian Research Council's Centre of Excellence for Climate System Science (http://climate-cms.unsw.wikispaces.net/Data). Global data were interpolated for each point to create the input data. Following your suggestion, we used data for which a 14-year spin-up was been performed for each point. Thank you for the suggestion.

"For the initial data on prognostic/diagnostic tracers in TOPAZ, we used the data provided by ARCCSS for use with MOM5 (http://climate-cms.unsw.wikispaces.net/Data). This initial tracer data was interpolated for each location, and a spin-up was applied over 14 years for use in the experiments."

6. "Page 8, Line 27: Just 4 years of spin up for a biogeochemical model is enough? Most of the applications with biogeochemical modeling are based in long spin up periods."

: Thank you for the good suggestion. As you suggested, we applied 14 years of spin-up to the initial data for each point and used it in the experiments. Please refer to the answer to specific comment #5.

7. "Page 8, Line 30: This similarity between GOTM-TOPAZ and observations is just on the first 40 meters for temperature. The difference in deeper regions must be discussed in this point."
:          We have revised the text as you suggested. The water temperature at point 107 as simulated by GOTM-TOPAZ showed a cold bias in the upper layer and a warm bias in the lower layer at a depth of around 120 m. We believe that this is an effect of the large-scale forcing (EKWC, ESIW) that is affecting this point. Thank you for the suggestion.

"Figure 3 shows the results of the GOTM-TOPAZ simulation and observational data (EN.4.2.1) as vertical distributions of the water column over time. The vertical distributions of salinity are well simulated and are comparable to the observations, although this could also be because relaxation was applied. The water temperature simulated by GOTM-TOPAZ showed a cold bias in the upper layer at a depth of around 120 m. This appears to be the effect of large-scale forcing (from the EKWC) that GOTM-TOPAZ could not resolve. Similar differences in water temperature also appeared at points 104 and 102 (Supplementary Figure 1)."

8. "Page 9, Line 5: Similarly to the latest comment, it is necessary to be clear in the text that this correspondence in seasonality between the model GOTM_TOPAZ and observation are just in the initial 40 meters."
:          Thank you for the suggestion. We have deleted this text and analyzed the reason for the difference in water temperature between GOTM-TOPAZ and the observations. Please refer to the response to specific comment #7 for more details.

9. "In Figure 3, there is no figure for observation to chlorophyll. In this case, I do not see a reason for this variable to be included in this figure."
:          We agree. We deleted the figure showing the results from MOM (water temperature and chlorophyll) from Fig. 3 and added a figure to show the difference in water temperature and salinity between the observations and the model. The revised figure is shown below.

[Figure]

**Figure 3: Comparison of the vertical distribution over time for water temperature [℃], salinity [psu] and the difference between the two (GOTM-TOPAZ results minus observational data) at point 107 for the 10-year period from 1999−2008.**

10. "Page 9, Line 13: These correlation coefficients are statically significant?"
:           We described the statistical significance levels of the correlations in this paper.

        "The mean chlorophyll concentration at depths of 0–20 m, as simulated by GOTM-TOPAZ and MOM, had similar inter-annual variabilities; their correlation coefficients versus the observational data were 0.53 and 0.60, respectively (Fig. 4a), which is statistically significant (p < 0.001)."

11. "Page 9, Line 6: It would be interesting here to discuss why the model GOTM-TOPAZ does not represent well the temperature in deeper regions, especially below 80 m. This discussion would be more interesting with the inclusion, in Figure 3, of a figure with the biases between models (MOM and GOTM-TOPAZ) and observations. Maybe this deficiency in the deeper regions is related with a short spin up period."
:           As you suggested, we used the results of 14 years of spin-up as the initial data for GOTM-TOPAZ. However, despite this, there was still a warm bias in the water temperature below a depth of 80 m (Fig. 3). We examined the difference between the observations and the simulated results for water temperature at points 107, 104, and 102 for a 22-year period 1987 to 2008. Fig. review 1 shows a cold region similar to the observations during the integral initial time at all three points; however, as time passed this region disappeared and the error increased. Therefore, we assumed that the difference in water temperature below 80 m between the model and the observations was caused by large-scale forcing in the ESIW. Of course, there will clearly be errors related to the spin-up, and we think that in the future it will be necessary to research the amplification of error as the integral time passes in a single-column model.

[Figure]

**Figure review 1: Comparison of the vertical spatial distribution over time for water temperature [℃] and difference (GOTM-TOPAZ minus obs.) for the period from 1987 to 2008; (a), (b), and (c) represent point 107, 104, and 102, respectively.**

12. "Page 9, line 30: The phrase "These results can be viewed as validating the gas flux equation reproduced in GOTM-TOPAZ" does not make much sense, once the correlation coefficient for GOTM-TOPAZ was worse than for MOM. Again, the correlation coefficients presented in Figure 5 are statically significant? In this paper, there was no evaluation of the fluxes. It is possible to evaluate the CO2 flux based on observational data, for instance, from SOCAT database."

: Following your suggestion, we have mentioned the statistical significance levels of all correlation coefficients, which we verified by comparing the $CO_2$ concentrations at the sea surface calculated by GOTM-TOPAZ with the SOCAT data. A scatterplot of the sea surface $CO_2$ concentrations calculated by GOTM-TOPAZ and SOCAT was added to the paper (Fig. 7). This figure shows that our model predicted $CO_2$ concentrations similar to the observations. Thank you for the useful comment.

"The mean chlorophyll concentration at depths of 0–20 m, as simulated by GOTM-TOPAZ and MOM, had similar inter-annual variabilities; their correlation coefficients versus the observational data were 0.53 and 0.60, respectively (Fig. 4a), which is statistically significant ($p < 0.001$)."

"the two models had a correlation coefficient of 0.59 ($p < 0.01$) and a similar inter-annual variability (Fig. 4b)."

"The sea surface dissolved oxygen levels simulated by GOTM-TOPAZ and MOM had correlation coefficients of 0.47 ($p < 0.001$) and 0.50 ($p < 0.001$), respectively, versus the observed data (Fig. 5a)."

"The GOTM-TOPAZ correlation coefficient versus the observed data was 0.31 ($p < 0.001$) for nitrogen, 0.16 ($p < 0.10$) for phosphorus, and 0.19 ($p < 0.05$) for silicon; these were lower than the correlation coefficients between MOM and the observed data (0.36, 0.24, and 0.33, respectively; $p < 0.001$). However, GOTM-TOPAZ seemed to depict the inter-annual variability of nutrients at the sea surface well (Fig. 5b–d)."

[Figure]

**Figure 7: Scatterplot of mean monthly sea surface CO2 concentration as observed by the Surface Ocean CO2 Atlas and modelled by GOTM-TOPAZ. The thin dotted lines around the 1-to-1 line represents ±1 and 2 µmol kg⁻¹.**

13. "Page 9, Line 21: The phrase: "In this paper, we have explained the major models that comprises GOTM-TOPAZ and the ocean biogeochemical process reproduced within the models" is not appropriated because you do not have made this on this paper. The model TOPAZ and the ocean biogeochemical process reproduced in this model was not explained on details on this paper. Actually, this explanation was not the main objective of this paper. To start the item discussion, I believe it would be more relevant to mention the main contributions of this paper, as a study about the development of a single-column ocean-biogeochemistry model."

:           Thank you for this correction. We revised the relevant section of the text. We have added a description of the fields of research that can use GOTM-TOPAZ to the Discussion section.

            "In this paper, we explain the major models that comprise GOTM-TOPAZ and the biological-physical feedback loop that they reproduce."
            " A variety of single-column ocean biogeochemical models have already been developed. However, GOTM-TOPAZ includes complex biogeochemical processes and models over 30 kinds of tracers; the other models, which have only simple structures, do not (Dunne et al., 2012b). Furthermore, GOTM-TOPAZ considers the gas transfer caused by changes in the atmosphere and the physical environment of the ocean, depicting the deposition of dissolved iron, lithogenic aluminosilicate, NH4, and NO3 due to aerosols. We believe that the sophistication of TOPAZ provides researchers with the opportunity to perform a variety of experiments.

   For example, the aerosol concentrations are continuously increasing in the over the East Asia region and are known to affect precipitation and atmospheric circulation. Thus, there is clearly a possibility that aerosols affect oceanic biogeochemical processes as deposition occurs in the ocean, and this cannot be ignored. A variety of numerical experiments are necessary to understand this process, but they are difficult to perform using 3D models due to limitations in computing resources. However, as previously noted, GOTM-TOPAZ is fast; as such, it is useful for understanding the biogeochemical changes that occur in the ocean when the concentration of aerosols or CO2 in the atmospheric change. In addition, recent studies have reported that the distribution of fisheries is changing due to changes in phytoplankton size structure, caused by upwelling intensity on the coast of the East/Japan Sea (Shin et al., 2017). Phytoplankton of TOPAZ is divided into two-types depending on their size, so it is expected to be useful in above mentioned research.
            In addition, GOTM-TOPAZ can be used in studies on feedback mechanisms in the biogeochemical and physical environment of the ocean."

14. "Page 10, Line 28: In this paper was not presented results about this sensitive experiments that are exemplified, how can you affirm that GOTM-TOPAZ will be good in this kind of applications? In the discussion topics, you should dedicate to discuss based on the results found in the paper."

:           We agree. We have deleted this text and added a description of the fields of research in which GOTM-TOPAZ can be applied, as mentioned in the answer to specific comment #13. Thank you for the useful suggestion.

15. "Finally, in the discussion, there was no evaluation about the upwelling representation. I believe that would be important to include in the paper the evaluation of the upwelling representation. How the module related to w-advection impacted the results? A comparison of the vertical movements reproduced by the model with observations would be interesting."

:           In the experiment at point 102, we prescribed the upwelling as decreasing linearly in the upward and downward directions at maximum value of 0.0000005 m/s, based on a depth of 100 m. The water temperature from GOTM-TOPAZ shown in Fig. review 2 demonstrates that a cold region exists due to upwelling at a depth of around 200 m. However, this cold region, which actually exists at point 102, is due to cold advection, and its mechanism is different from the upwelling experiment. Nonetheless, we performed experiments to verify the implementation of upwelling in the model. The

mean chlorophyll concentration at depths of 20−80 m show that there is a rapid increase in upwelling during winter (Fig. review 3). Because of the effect of this upwelling, the nutrients below a depth of

200 m were supplied to the upper layer during the previous period. The supplied nutrients are consumed and thus have an effect on the increase in chlorophyll concentration around at 20−80 m in the winter (Fig. review 4). The effect of the upwelling is also seen in the vertical profile of dissolved oxygen. Fig. review 4 shows that the middle layer of seawater, which is deeper than 300 m (where there is little dissolved oxygen), was supplied to the upper layer, and the concentration of dissolved oxygen below a depth of 150 m decreased sharply. We still do not have adequate data to implement upwelling that is similar to reality. In the future, we plan to collect observational data related to this and perform a study on upwelling on the eastern coast and changes in the ocean biogeochemical environment.

[Figure]

**Figure review 2: Comparison of the vertical distribution over time for water temperature [℃] (a), salinity [psu] (b) and the difference between the two (GOTM-TOPAZ minus obs.) at point 102 for the 10-year period from 1999–2008. The upwelling is prescribed to the GOTM-TOPAZ.**

[Figure]

**Figure review 3: Chlorophyll time series from GOTM-TOPAZ (red lines) and upwelling case (blue lines) at point 102 for the 10-year period from 1999–2008. Chlorophyll concentration is averaged between 20 to 80m depth. The upwelling is prescribed to the GOTM-TOPAZ.**

[Figure]

**Figure review 4: Vertical profile from the GOTM-TOPAZ (red dots) and upwelling case (blue dots) with respect to dissolved oxygen, nitrogen, phosphorus, and silicon averaged from 1999 to 2008; (a) for February; (b) for August; and (c) annually. The shaded area represents 1 sigma. In this figure, nitrogen, phosphorus, and silicon include NO3, PO4, and SIO4, respectively.**

---

## Author Comment (AC3) · 30 Nov 2018

The comment was uploaded in the form of a supplement:
https://www.geosci-model-dev-discuss.net/gmd-2018-200/gmd-2018-200-AC3-supplement.pdf

---

## Author Comment (AC4) · 30 Nov 2018

**Supplementary Figures**

We verified the modeling performance of GOTM-TOPAZ for point 104 (131.3° E, 37.1° N) and point 102 (130.6° E, 36.1° N) in addition to point 107 (130° E, 38.0° N). The water temperatures at the former two points, like that at point 107,

5    exhibited a cold bias in the upper layer and a warm bias in the lower layer at a depth of around 120 m (Supplementary Figure 1). This demonstrates influence from the East Korea Warm Current (EKWC) and the East Sea Intermediate Water (ESIW) that GOTM-TOPAZ could not resolve. In terms of the chlorophyll at a depth of 80 m, GOTM-TOPAZ showed an inter-annual variability similar to those from the results from observational data and MOM at both points 104 and 102 (Supplementary Figure 2, 5). Furthermore, the sea surface dissolved oxygen, nitrogen, and phosphorus concentrations

10    simulated by GOTM-TOPAZ at those two points had correlation coefficients that were slightly lower or similar to those of results from MOM (Supplementary Figures 3, 6). The vertical distributions of dissolved oxygen and nutrients at points 104 and 102 showed patterns similar to that at point 107 when simulated by GOTM-TOPAZ (Supplementary Figures 4, 7).

[Figure]

**Supplementary Figure 1: Comparison of the vertical distribution over time for water temperature [℃], salinity [psu] and the difference between the two (GOTM-TOPAZ minus observational data) at points (a) 104 and (b) 102 for the 10-year period 1999–2008.**

[Figure]

**Supplementary Figure 2: Chlorophyll anomaly time series and correlation values for observational data (black lines), MOM5_SIS_TOPAZ results (blue lines), and GOTM-TOPAZ results (red lines) at point 104 for the 10-year period 1999−2008; (a) the mean value at depths ≥ 20 m and the correlations between the observations and each model; (b) mean values at depths of 20–80 m and the correlation between the two models.**

[Figure]

**Supplementary Figure 3: Anomaly time series and correlation values from observational data (black lines), MOM results (blue lines), and GOTM-TOPAZ results (red lines) for concentrations of (a) dissolved oxygen, (b) nitrogen, (c) phosphorus, and (d) silicon at point 104 for the 10-year period 1999−2008; in this figure, nitrogen, phosphorus, and silicon include $NO_3$, $PO_4$, and $SIO_4$, respectively.**

[Figure]

**Supplementary Figure 4: Vertical profile from observational data (black dots) and GOTM-TOPAZ results (red dots) at point 104 for concentrations of dissolved oxygen, nitrogen, phosphorus, and silicon averaged from 1999–2008 (a) for February; (b) for August; and (c) annually. The shaded areas represent 1 sigma. In this figure, nitrogen, phosphorus, and silicon include $NO_3$, $PO_4$, and $SIO_4$, respectively.**

[Figure]

**Supplementary Figure 5: Chlorophyll anomaly time series and correlation values for observational data (black lines), MOM5_SIS_TOPAZ results (blue lines), and GOTM-TOPAZ results (red lines) at point 102 for the 10-year period 1999−2008; (a) the mean value at depths ≥ 20 m and the correlations between the observations and each model; (b) mean values at depths of 20–80 m and the correlation between the two models.**

[Figure]

**Supplementary Figure 6: Anomaly time series and correlation values from observational data (black lines), MOM results (blue lines), and GOTM-TOPAZ results (red lines) for concentrations of (a) dissolved oxygen, (b) nitrogen, (c) phosphorus, and (d) silicon at point 102 for the 10-year period 1999–2008; in this figure, nitrogen, phosphorus, and silicon include NO3, PO4, and SIO4, respectively.**

[Figure]

**Supplementary Figure 7:** Vertical profile from observational data (black dots) and GOTM-TOPAZ results (red dots) at point 102 for concentrations of dissolved oxygen, nitrogen, phosphorus, and silicon averaged from 1999–2008 (a) for February; (b) for August; and (c) annually. The shaded areas represent 1 sigma. In this figure, nitrogen, phosphorus, and silicon include $NO_3$, $PO_4$, and $SIO_4$, respectively.

---

## Author Response (AR2)

**Reply to Editor's Comments and Suggestions**

Manuscript number: gmd-2018-200
Title: A single-column ocean-biogeochemistry model (GOTM-TOPAZ) version 1.0

We appreciate your considered comments and suggestions, which have proven very helpful in improving our manuscript as well as very valuable in guiding our future research. We have made some revisions to the manuscript in accordance with your comments. The revised portions of the manuscript are marked in red, while our detailed responses below are given in blue.

We greatly appreciate the time and effort you have given to assessing our work and, once again, we thank you very much for your kind comments and suggestions.

**Editor**

[General Comments]

Thank you also for your revised manuscript and response to reviewers. While I feel that in the context of a GDM paper, many of the points have now been addressed or are not of major concern, there are a small number of further changes which I would like to see before I can make a final decision (potentially with the help of the reviewers).

:        Thank you for your meaningful review and comments. We addressed each of the specific comments and responded to them individually below. We hope that the revised manuscript is now ready for publication.

[Specific Comments]

1. "The main issue which I would like you to consider further is the the more complete integration of the additional site into main manuscript. I share the reviewers concerns that the main site you have chosen has not been demonstrated to be well suited for 1D modelling, and as such it is hard for the reader to get a good feeling for the true skill of the model. Assuming that you can adequately justify that the two additional sites are suitable locations for 1D modelling, please can these be given equal weight to the 1st site in the main manuscript. I appreciate that this might require some consolidation and reorganisation of figures."

:        Thank you for the useful comment. The majority of observation points in the sea off the Korean Peninsula have been located on the paths of either the Tsushima Warm Current or the East Korea Warm Current. Besides, there are only a few points at which biogeochemical variables have been observed over a long period of time (at least 10 years). This provided us with a constraint for selecting points from which observed values that were necessary to verify the model could be obtained.

We selected the observation points by considering the continuity and quality of the ocean biogeochemical observation data. Point 107 is located where the North Korea Cold Current and the East Korea Warm Current meet. This area is biogeochemically important and has been actively studied with respect to the variations in the main fish species and catch according to phytoplankton characteristics (size or toxic/non-toxic) (Joo et al., 2014; Shin et al., 2017). The other two points 104 and 102 are located on the path of the East Korea Warm Current. Warm eddies are also observed at these points. Of course, as these points are greatly affected by the ocean current, extra attention needs to be paid when verifying a single column model (SCM).

Despite such a disadvantage, if GOTM-TOPAZ produces meaningful results, the expendability of the model (that is, the applicability to various observational points) will be proven. We expect that

out model will be applied to many ocean biogeochemical investigations and be suitably tuned for each area. Following the editor's request, we have added analysis results about the additional two points (104, 102) in the manuscript.

Joo, H. T., Park, J. W., Son, S. H., Noh, J.–H., Jeong, J.-Y., Kwak, J. H., Saux-Picart, S., Choi, J. H., Kang, C.-K., and Lee, S. H.: Long-term annual primary production in the Ulleung Basin as a biological hot spot in the East/Japan Sea, J. Geophys. Res. Oceans, 119, 3002–3011, doi:10.1002/2014JC009862, 2014.

Shin, J.-W., Park, J., Choi, J.-G., Jo, Y.-H., Kang, J. J., Joo, H. T., and Lee, S. H.: Variability of phytoplankton size structure in response to changes in coastal upwelling intensity in the southwestern East Sea, J. Geophys. Res. Oceans, 122, 10, 262–10, 274, doi:10.1002/2017JC013467, 2017.

2. "Finally, I appreciate the changes you have made to avoid over-representing the skill of the model in the text. I would like to see this go further. For example, where you have looked at profiles, there are places depths at which the absolute values of variables are correct, but it is on a background of a profile which is of the wrong shape. Given the nature of a 1D model, where processes are occurring only vertically, my assumption in most cases is that if (for example) the middle part of the profile matches the absolute values of the observations, but the top and bottom of the water column are wrong, this has to arise through the cancellation of errors rather than inherent skill. Please can you therefore further revise the validation component of the text. The important thing for GMD is not that the model is highly skilful, rather that it is impartially evaluated so that other potential users can simply understand the model and its performance."

: We agree with the editor's comment that the similarity of the biogeochemical variables between our model simulation and the observational results might be attributed to the cancellation of errors (irrespective of the performance of the model). Accordingly, following the editor's suggestion, we have revised the validation component. As for dissolved oxygen, the bias was larger in summer, and there was also a bias in the deep sea ($< 250$ m). For this reason, we analyzed the magnitude of the source and sink terms instead of vertical diffusion.

The production of dissolved oxygen is attributable to nitrate, ammonia, and nitrogen fixation (caused by phytoplankton), while the loss occurs during the production of $NH_4$ from non-sinking particles, sinking particles, and nitrification (Dunne et al., 2012b). As shown in Figures 8, 9, and 10, the model excessively simulated dissolved oxygen in the surface layer ($< 60$ m) in the summer. This seems to be because the photosynthesis of phytoplankton is dominant. In addition, our model tended to underestimate dissolved oxygen in the deep sea ($> 250$ m). This error was (significantly) reduced by using the observations for the initial data. We found that our model was sensitive to source/sink terms in the surface layer and initial values in the deep sea. Other variables showed a similar result. Accordingly, model users need to consider this characteristic of GOTM-TOPAZ while conducting an experiment. We have added this point to the Abstract, Results, and Discussion.